# Beyond Binary Preferences: A Principled Framework for Reward Modeling with Ordinal Feedback

**Amirhossein Afsharrad**[1,5],[*] **Ruida Zhou**[2]**, Luca Viano**[3]**,**
**Sanjay Lall**[1]**, Mohammad Ghavamzadeh**[4]

[1]Stanford University, [2]Amazon AGI, [3]EPFL, [4]Qualcomm AI Research, [5]Aktus AI

## Abstract

Reward modeling is crucial for aligning large language models with human preferences, yet current approaches lack a principled mathematical framework for leveraging ordinal preference data. When human annotators provide graded preferences on a Likert scale (e.g., significantly better, better, slightly better, negligibly better), existing methods typically apply ad-hoc heuristics, such as margin terms or scaling factors, to loss functions derived from binary preference models like Bradley-Terry. These approaches lack an underlying mathematical model for how ordinal preference data is generated. We present a theoretically grounded framework that formulates reward modeling with Likert scale preferences as a discrete ordinal regression problem. We derive two loss functions from this formulation: a negative log-likelihood loss and an all-threshold loss, both of which learn threshold parameters that naturally capture the ordinal structure of preferences. Unlike existing heuristic methods that manually specify fixed margins or scaling weights, our approach learns these parameters directly from data within a coherent probabilistic framework. Experimental results on multiple benchmarks demonstrate that our ordinal regression approach consistently achieves competitive or superior performance compared to existing heuristic methods across diverse evaluation categories including chat, reasoning, and safety tasks. Our work provides the first principled mathematical framework for incorporating Likert scale preferences into reward model training, moving beyond ad-hoc modifications of binary preference models to enable more effective utilization of fine-grained human feedback.

## 1 Introduction

Human feedback drives modern language model alignment. Through reinforcement learning from human feedback (RLHF) (Christiano et al., 2017; Ziegler et al., 2019; Stiennon et al., 2020; Bai et al., 2022; Ouyang et al., 2022) and Direct Preference Optimization (DPO) style algorithms (Rafailov et al., 2023; Azar et al., 2023; Ethayarajh et al., 2024; Xu et al., 2024; Hong et al., 2024; Meng et al., 2024), models learn to generate outputs that humans prefer. Both approaches fundamentally rely on the Bradley-Terry model (Bradley and Terry, 1952), which connects to human preferences through a reward function—explicitly in RLHF and implicitly in DPO. This model assumes that the probability of preferring one response over another increases monotonically with the difference between their reward values. Crucially, Bradley-Terry was designed for binary comparisons: response A is either preferred or dispreferred to response B. This binary foundation means that when preference data contains richer information, there is no systematic way to leverage it within these frameworks. Yet increasingly, preference datasets capture more than binary choices. When annotators compare responses, they often specify not just which one is better but by how much—assigning ordinal ratings like "slightly better," "moderately better," or "significantly better" on Likert scales (Wang et al., 2024c; 2025). This ordinal structure contains substantially more signal than binary labels, but current methods, constrained by their binary foundations, largely ignore it.

---

[*]Corresponding author: `afsharrad@stanford.edu`

Current attempts to leverage ordinal preferences mainly rely on heuristics, such as adding margins between preference levels (Touvron et al., 2023), scaling losses by preference strength (Wang et al., 2024a), or treating ordinal levels as soft probability labels (Gunter et al., 2024; Liu et al., 2024a). These approaches share two fundamental problems. First, they lack any mathematical model of how humans assign ordinal preferences—they simply modify the binary Bradley-Terry loss with ad-hoc terms chosen through intuition rather than principle. Second, they require practitioners to manually tune hyper-parameters that have no clear interpretation: what margin should separate "slightly better" from "significantly better"? Should a strong preference contribute twice or thrice the gradient of a weak one? Moreover, these hyper-parameters must be retuned whenever the number or definition of preference levels changes, making the approaches brittle and dataset-specific.

We reframe reward modeling with ordinal feedback as a discrete ordinal regression problem—a well-established statistical framework for ordered categorical data (McCullagh, 1980; Winship and Mare, 1984). The machine learning community has developed extensive methodology for such problems (Chu and Keerthi, 2005; 2007; Shashua and Levin, 2002; Rennie and Srebro, 2005; Srebro et al., 2004), yet this machinery has not been applied to preference learning. Our approach models the relationship between reward differences and preference levels through learned thresholds that partition the continuous reward space. Rather than manually specifying margins or scaling factors, we derive principled loss functions—both probabilistic based on cumulative link models and margin-based inspired by structured prediction—that learn all parameters directly from data.

Our work demonstrates that treating preferences as inherently ordinal yields consistent improvements across benchmarks and architectures. Beyond standard binary accuracy metrics, our methods accurately predict preference *strength*, achieving approximately 85% accuracy within one ordinal level on held-out data. The learned threshold structures provide interpretable insights into how annotators distinguish between preference levels, revealing patterns that manual hyper-parameter tuning could never discover. As preference collection evolves toward richer annotation schemes—confidence scores, multi-aspect ratings, and other forms of structured feedback—principled frameworks become essential. Our ordinal regression approach provides a mathematical foundation that naturally extends to handle these diverse forms of feedback. The core insight is simple: human preferences often carry ordinal information, and it would be much better if our methods leverage this structure rather than forcing it through binary models with heuristic patches.

## 2 PRELIMINARIES

**Notations.** For any positive integer $K$, we denote the set $\{1, 2, \ldots, K\}$ by $[K]$ and the set $\{-K, -K+1, \ldots, -1\}$ by $[-K]$. Throughout this paper, we use $\sigma(t) = \frac{1}{1+e^{-t}}$ to denote the sigmoid (logistic) function.

### 2.1 DISCRETE ORDINAL REGRESSION

Ordinal regression addresses prediction problems where the target variable takes values from a discrete ordered set. Unlike standard classification, where categories are unrelated, or regression, where outputs are continuous, ordinal regression must respect the inherent ordering among categories while maintaining their discrete nature. Common applications include rating systems (1-5 stars), survey responses (strongly disagree to strongly agree), and severity levels (mild, moderate, severe).

Given a training dataset $\mathcal{D} = \{(x_i, z_i)\}_{i=1}^{n}$ where $x_i \in \mathbb{R}^d$ is a feature vector and $z_i \in [K]$ is an ordinal label, the goal is to learn a model that predicts the correct ordinal level for new inputs. Most ordinal regression methods model the ordinal structure through a latent continuous variable that is subsequently discretized. Specifically, these methods learn a scoring function $s_\phi : \mathbb{R}^d \to \mathbb{R}$ parameterized by $\phi$, which maps inputs to a one-dimensional latent space, along with a set of ordered thresholds $-\infty = \zeta_0 < \zeta_1 < \cdots < \zeta_{K-1} < \zeta_K = +\infty$ that partition the latent space into $K$ intervals. The thresholds create a correspondence between latent scores and ordinal levels: higher scores correspond to higher ordinal categories. This framework reduces to binary classification when $K = 2$, with a single threshold (typically zero) separating the two classes.

Given this latent variable framework, there are two fundamentally different approaches to learning the parameters $(\phi, \zeta)$: probabilistic modeling via generalized linear models and direct loss minimization through margin-based methods.

**Probabilistic Modeling via Generalized Linear Models (GLMs).** The probabilistic approach explicitly models the conditional distribution $\mathbb{P}(z|x)$ through cumulative probabilities. The ordinal regression model assumes

$$\mathbb{P}(z \leq k \mid x) = f(\zeta_k - s_\phi(x)), \tag{1}$$

where $f : \mathbb{R} \to [0, 1]$ is a cumulative distribution function. The most common choice is $f(t) = \sigma(t)$, yielding the ordered logit model. An alternative is $f(t) = \Phi(t)$ (the standard normal CDF), yielding the ordered probit model. This cumulative model naturally ensures that $\mathbb{P}(z \leq k) \geq \mathbb{P}(z \leq k - 1)$ for all $k$. Thus, the probability of observing exactly level $k$ is obtained by

$$\mathbb{P}(z = k \mid x) = f(\zeta_k - s_\phi(x)) - f(\zeta_{k-1} - s_\phi(x)). \tag{2}$$

For the ordered logit model with $f = \sigma$, maximum likelihood estimation leads to minimizing the negative log-likelihood

$$\mathcal{L}_{\text{NLL}}(s_\phi, \zeta; z) = - \log \left[ \sigma(\zeta_z - s_\phi(x)) - \sigma(\zeta_{z-1} - s_\phi(x)) \right], \tag{3}$$

where we have simplified the notation slightly by dropping the explicit dependence on $x$ in the loss function. This loss penalizes the model for assigning low probability mass to the interval containing the true label. Note that this probabilistic formulation is specific to the GLM approach and provides a complete model for $\mathbb{P}(z|x)$.

**Margin-Based Direct Loss Minimization.** As an alternative to probabilistic modeling, Rennie and Srebro (2005) proposed methods that directly penalize threshold violations without explicitly modeling probabilities. These approaches, inspired by large-margin methods in machine learning, focus on ensuring the latent score falls within the correct interval. The framework uses a margin penalty function $g : \mathbb{R} \to \mathbb{R}_+$ where $g(u)$ measures the cost of violating a margin constraint by amount $u$. Common choices of penalty functions include logistic $g(u) = - \log \sigma(u)$, hinge $g(u) = \max(0, 1 - u)$, and exponential $g(u) = e^{-u}$. Two loss functions emerge from this framework. The **Immediate-Threshold (IT) Loss** only considers the boundaries of the target interval,

$$\mathcal{L}_{\text{IT}}(s_\phi, \zeta; z) = g(s_\phi(x) - \zeta_{z-1}) + g(\zeta_z - s_\phi(x)). \tag{4}$$

This loss encourages $s_\phi(x)$ to lie within $(\zeta_{z-1}, \zeta_z)$ but treats all misclassifications equally, regardless of severity. The **All-Threshold (AT) Loss** accumulates penalties from all threshold violations,

$$\mathcal{L}_{\text{AT}}(s_\phi, \zeta; z) = \sum_{l=1}^{K-1} g\big(\nu(l; z) \cdot (\zeta_l - s_\phi(x))\big), \tag{5}$$

where $\nu(l; z) = -1$ if $l < z$ and $\nu(l; z) = +1$ if $l \geq z$. This formulation ensures that we penalize $s_\phi(x) < \zeta_l$ when the true label $z > l$, and penalize $s_\phi(x) > \zeta_l$ when $z \leq l$. The AT loss increases with the degree of misclassification, making it more sensitive to large errors than the IT loss. For the logistic margin penalty $g(u) = - \log \sigma(u)$, these losses can be written as

$$\mathcal{L}_{\text{IT}}(s_\phi, \zeta; z) = - \log \sigma(s_\phi(x) - \zeta_{z-1}) - \log \sigma(\zeta_z - s_\phi(x)), \tag{6}$$

$$\mathcal{L}_{\text{AT}}(s_\phi, \zeta; z) = \sum_{l=1}^{K-1} - \log \sigma\big(\nu(l; z) \cdot (\zeta_l - s_\phi(x))\big). \tag{7}$$

The key distinction between these two approaches is that the probabilistic method models the full conditional distribution $\mathbb{P}(z|x)$ and uses maximum likelihood estimation, while the margin-based methods focus on correct classification through direct optimization without probabilistic interpretation. The margin-based losses are often computationally simpler, particularly for large-scale problems, as noted by Rennie and Srebro (2005) who observed that "even just evaluating the likelihood of a predictor is not straight-forward" in ordered logit and probit models.

In subsequent sections, we will employ both the NLL loss from the probabilistic approach (Eq. 3) and the AT loss from the margin-based approach (Eq. 7). We exclude the IT loss since the empirical results in Rennie and Srebro (2005) demonstrate its inferior performance compared to the AT loss.

## 2.2 REWARD MODELING

Reward modeling followed by policy optimization using a deep reinforcement learning algorithm is a main approach to post-training LLMs, often referred to as RLHF. A reward model (RM) is trained on a dataset of comparisons between two responses $(y, y')$ to the same input prompt $x$. Learning an RM consists of training a binary classifier to discriminate between the preferred and dispreferred responses using a logistic regression loss. A popular choice for the classifier is the Bradley-Terry (BT) model. The BT model stipulates that the human preference distribution $p^\star$ can be written as

$$p^\star(y \succ y'|x) = \sigma\big(r^\star(x, y) - r^\star(x, y')\big), \tag{8}$$

where $r^\star(x, y)$ is the (latent) reward function of the annotator for prompt $x$ and response $y$.

Under the BT preference model and given a training set $\mathcal{D} = \{x_i, y_{w,i}, y_{l,i}\}_{i=1}^n$ in which $y_w$ and $y_l$ denote preferred and dispreferred responses for the prompt $x$, one can learn the RM by minimizing the negative log-likelihood loss

$$\mathcal{L}_{\text{RM}}(r_\phi; \mathcal{D}) = -\mathbb{E}_{(x, y_w, y_l) \sim \mathcal{D}}\big[\log \sigma\big(r_\phi(x, y_w) - r_\phi(x, y_l)\big)\big]. \tag{9}$$

This loss encourages the reward model to assign higher scores to preferred responses compared to dispreferred ones. The learned reward model $r_\phi$ can then be used to provide reward signals for RL algorithms that optimize the language model policy. While we focus on reward modeling in this work, the techniques we develop can be adapted to Direct Preference Optimization (DPO) (Rafailov et al., 2023) style algorithms, as we will discuss in Appendix A.

## 3 ORDINAL REGRESSION FOR REWARD MODELING

In this section, we first formulate the problem of reward modeling using ordinal preference data. We then provide a brief overview of the existing heuristic approaches to this problem. Finally, we present our approach which is base on the ordinal regression framework and present our algorithms for learning a reward model in this setting.

### 3.1 PROBLEM FORMULATION

We consider the problem of learning reward models (RMs) from preference data with ordinal feedback. Unlike standard preference learning where annotators provide binary comparisons, we consider the more informative setting where annotators indicate not only which response is preferred but also the degree of their preference. More specifically, for each input prompt $x$ and two candidate responses $y$ and $y'$, the annotator provides feedback on a Likert scale indicating how much better one response is compared to the other. We denote $y \succ_k y'$ to indicate that $y$ is compared to $y'$ at level $k$, where $k \in [-K] \cup \{0\} \cup [K]$. When $k \in [K]$, this means $y$ is preferred to $y'$ at level $k$, with higher values indicating stronger preference. When $k \in [-K]$, this means $y$ is dispreferred relative to $y'$ at level $|k|$. The case $y \succ_0 y'$ indicates that the annotator finds the two responses approximately equal or is unsure which one is better. This results in a dataset of the form $\mathcal{D} = \{(x_i, y_i, y_i', z_i)\}_{i=1}^n$, where each tuple contains a prompt, two responses, and the ordinal preference $z_i \in [-K] \cup \{0\} \cup [K]$ such that $y_i \succ_{z_i} y_i'$. In practice (e.g., Wang et al. 2024b), annotators might use labels such as "significantly better", "better", "slightly better", and "negligibly better", which correspond to different values of $|z_i|$ depending on which response is preferred.

The goal is to use $\mathcal{D}$ and learn a reward model $r_\phi(x, y)$ parameterized by $\phi$ that can score individual responses in a way that respects the ordinal preference structure. Following the BT model, the natural predictor for this problem takes the form of reward differences,

$$s_\phi(x, y, y') = r_\phi(x, y) - r_\phi(x, y'). \tag{10}$$

The challenge is to learn $r_\phi$ such that the reward difference $s_\phi(x, y, y')$ not only has the correct sign (indicating which response is preferred) but also the correct magnitude (reflecting the degree of preference expressed by $z$).

### 3.2 EXISTING HEURISTIC APPROACHES AND THEIR LIMITATIONS

Current approaches to learning from ordinal preference data modify the standard BT loss in various ad-hoc ways to incorporate the preference strength information.

**Margin Bradley-Terry.** Proposed in Llama-2 (Touvron et al., 2023), this approach adds a margin term to the standard BT loss,

$$\mathcal{L}_{\text{MBT}}(r_\phi; x, y, y', z) = -\log \sigma \big( r_\phi(x, y) - r_\phi(x, y') - m(z) \big), \tag{11}$$

where the margin $m : \{-K, \ldots, K\} \to \mathbb{R}$ is a manually specified function of the preference rating. For instance, Touvron et al. (2023) experimented with $m(k) = (3, 2, 1, 0)$ for $k \in \{$significantly better, better, slightly better, negligibly better$\}$.

**Scaled Bradley-Terry.** In HelpSteer2, Wang et al. (2024a) proposed scaling the loss itself by the preference strength,

$$\mathcal{L}_{\text{SBT}}(r_\phi; x, y, y', z) = -m(z) \cdot \log \sigma \big( r_\phi(x, y) - r_\phi(x, y') \big), \tag{12}$$

where $m(k) = (3, 2, 1)$ for different preference levels.

**Soft Label.** Gunter et al. (2024) interpreted ordinal feedback as soft labels,

$$\mathcal{L}_{\text{SL}}(r_\phi; x, y, y', z) = -p(z) \cdot \log \sigma \big( r_\phi(x, y) - r_\phi(x, y') \big) - \big( 1 - p(z) \big) \cdot \log \sigma \big( r_\phi(x, y') - r_\phi(x, y) \big), \tag{13}$$

where $p(z)$ represents an arbitrary probability assignment, such as $p(k) = (0.95, 0.85, 0.75, 0.65)$ for different preference levels.

These approaches share two fundamental limitations. First, they lack a coherent mathematical model of ordinal preferences. While the BT model provides a principled probabilistic framework for binary preferences, i.e., $p(y \succ y' | x) = \sigma(r(x, y) - r(x, y'))$, these extensions have no corresponding model that explains how humans assign ordinal labels. The modifications to the loss function are arbitrary and it is unclear what assumptions about human preference behavior would lead to these particular formulations. Second, they require manual specification of their hyper-parameters $m(\cdot)$ or $p(\cdot)$ with no principled method for choosing their values or learning them from data. In contrast, ordinal regression provides a well-established statistical framework where the loss functions arise naturally from explicit modeling assumptions and all parameters are learned from data.

### 3.3 A Principled Ordinal Regression Framework

We propose formulating reward modeling with ordinal feedback as a discrete ordinal regression problem. Unlike the heuristic approaches, ordinal regression provides principled methods where the relationship between reward differences and preference levels is learned from data rather than manually specified. To extend ordinal regression to preference learning, we introduce $2K$ thresholds $\zeta_{-K}, \ldots, \zeta_{-1}, \zeta_1, \ldots, \zeta_K$ that partition the real line into $2K + 1$ segments. For convenience, we denote $\zeta_{-(K+1)} = -\infty$ and $\zeta_{K+1} = +\infty$. Note that we deliberately exclude $\zeta_0$; the segment between $\zeta_{-1}$ and $\zeta_1$ corresponds to the preference level $z = 0$ (approximately equal responses).

Given that we are modeling preferences, the predictor values for our ordinal regression problem take the form described in (10). These thresholds serve different roles depending on the modeling paradigm we adopt. As we describe in the following sections, the probabilistic approach uses thresholds to define cumulative probabilities for different preference levels, while the margin-based approach uses them as boundaries that the predictor should respect. Both approaches learn the thresholds from data, eliminating the arbitrary hyper-parameter choices required by the heuristic methods. The key insight is that ordinal regression provides established frameworks—both statistical and optimization-based—for learning from ordered categorical data. We adapt these frameworks to the specific structure of preference learning, where the ordinal variable represents the degree and direction of preference between two responses.

#### 3.3.1 Loss Functions

We adapt two approaches from discrete ordinal regression to our preference learning setting, each representing a different paradigm for utilizing ordinal information.

**Probabilistic Approach: Negative Log-Likelihood.** The probabilistic approach assumes that human annotators follow an ordered logit model when assigning preference levels. Given reward dif-

ference $s_\phi(x, y, y')$, the probability of observing preference level $z$ is modeled as

$$p(y \succ_z y' \mid x) = \begin{cases} \sigma(\zeta_z - s_\phi(x, y, y')) - \sigma(\zeta_{z-1} - s_\phi(x, y, y')) & \text{if } z \in [-K], \\ \sigma(\zeta_{z+1} - s_\phi(x, y, y')) - \sigma(\zeta_z - s_\phi(x, y, y')) & \text{if } z \in [K], \\ \sigma(\zeta_1 - s_\phi(x, y, y')) - \sigma(\zeta_{-1} - s_\phi(x, y, y')) & \text{if } z = 0. \end{cases} \quad (14)$$

This formulation is a direct application of the ordered logit model from (2) to our preference learning setting, where the predictor value is the reward difference $s_\phi(x, y, y')$. The model ensures that probabilities sum to one across all preference levels.

Maximizing the likelihood of the observed preferences leads to the negative log-likelihood loss,

$$\mathcal{L}_{\text{NLL}}(r_\phi, \zeta; x, y, y', z) = -\log p(y \succ_z y' \mid x), \quad (15)$$

where $p(y \succ_z y' \mid x)$ is given by (14). This loss has a clear interpretation: it penalizes the model for assigning low probability to the observed preference level.

**Margin-Based Approach (All-Threshold Loss).** The margin-based approach does not assume a specific probabilistic model of human behavior. Instead, inspired by large-margin methods in machine learning, it directly penalizes violations of the ordinal structure. The AT loss accumulates penalties for the reward difference being on the wrong side of each threshold,

$$\mathcal{L}_{\text{AT}}(r_\phi, \zeta; x, y, y', z) = \sum_{l \in [-K] \cup [K]} -\log \sigma\big(\nu(l; z) \cdot (\zeta_l - s_\phi(x, y, y'))\big), \quad (16)$$

where $\nu(l; z) = -1$ if $l < z$ and $\nu(l; z) = +1$ if $l \geq z$. This formulation encourages $s_\phi(x, y, y')$ to be greater than all thresholds $\zeta_l$ with $l < z$ and less than all thresholds $\zeta_l$ with $l \geq z$, with logarithmic penalties for violations.

**Regularization for Stable Optimization.** The simultaneous learning of reward parameters $\phi$ and thresholds $\zeta$ introduces a fundamental challenge in optimization. As we formalize below, without regularization, the optimization problem admits unbounded solutions.

**Theorem 3.1** (Unbounded Solution Set). *Consider either the NLL loss $\mathcal{L}_{NLL}$ or the AT loss $\mathcal{L}_{AT}$ without regularization. Suppose there exists a solution $(r_\phi^*, \zeta^*)$ such that all training examples are correctly ordered, i.e., $s_i^* \in (\zeta_{z_i-1}^*, \zeta_{z_i}^*)$ for all $(x_i, y_i, y_i', z_i) \in \mathcal{D}$, where $s_i^* = r_\phi^*(x_i, y_i) - r_\phi^*(x_i, y_i')$. Then for any $c > 1$, the scaled solution $(cr_\phi^*, c\zeta^*)$ achieves strictly lower loss, and thus, $\lim_{c \to \infty} \sum_i \mathcal{L}(cr_\phi^*, c\zeta^*; x_i, y_i, y_i', z_i) = 0$. Consequently, the unregularized optimization problem has no finite optimal solution.*

The proof, reported in Appendix B, shows that correctly classified examples contribute vanishing loss as the scale increases, while misclassified examples incur linearly growing penalties. This creates optimization pathologies: gradient descent can drive thresholds to arbitrarily large values while maintaining or reducing the loss, leading to numerical instability and poor model calibration.

To address this issue, we add an $L_2$-regularization term on the thresholds, yielding the following regularized objective:

$$\min_{\phi, \zeta \in \mathcal{C}} \sum_{(x, y, y', z) \in \mathcal{D}} \mathcal{L}(r_\phi, \zeta; x, y, y', z) + \lambda \|\zeta\|_2^2, \quad (17)$$

where $\mathcal{C} = \{\zeta \in \mathbb{R}^{2K} : \zeta_{-K} < \ldots < \zeta_{-1} < \zeta_1 < \ldots < \zeta_K\}$ and $\mathcal{L}$ can be either $\mathcal{L}_{\text{NLL}}$ or $\mathcal{L}_{\text{AT}}$. The regularization term prevents unbounded growth of thresholds and ensures the existence of a finite optimal solution.

### 3.3.2 SYMMETRIC AND ASYMMETRIC MODELS

An important modeling choice in our framework concerns whether to impose symmetry constraints on the thresholds. This choice is orthogonal to the selection of loss function and can be applied with either the probabilistic or margin-based approach.

**Symmetric Model.** A natural assumption is that the strength of preference should be symmetric: preferring $y$ over $y'$ at level $k$ should correspond to the same strength as preferring $y'$ over $y$ at level $-k$. Formally, this translates to imposing the constraint $\zeta_{-k} = -\zeta_k$, $\forall k \in [K]$. Under the probabilistic model, this symmetry constraint has the following theoretical justification.

**Theorem 3.2** (Threshold Symmetry under Ordered Logit). *Consider the ordered logit model in* (14). *If the preference data satisfies the symmetry property that for all $k \in [K]$ and all reward differences $r \in \mathbb{R}$,*

$$\mathbb{P}(y \succ_k y' \mid s_\phi(x, y, y') = r) = \mathbb{P}(y' \succ_{-k} y \mid s_\phi(x', y, y') = r), \tag{18}$$

*then the thresholds must satisfy $\zeta_{-k} = -\zeta_k$ for all $k \in [K]$.*

The proof, reported in Appendix C, shows that the symmetry in human preferences necessarily implies symmetry in the thresholds. While this theorem provides theoretical justification under the probabilistic model, the symmetry constraint can also be imposed with the margin-based approach as a reasonable structural assumption. The symmetric model reduces the number of threshold parameters from $2K$ to $K$, improving computational efficiency and reducing overfitting risk.

**Asymmetric Model.** Human preferences often exhibit asymmetries due to various cognitive biases. For instance, annotators might be more conservative when strongly endorsing one response over another (high positive $k$) than when strongly criticizing it (high negative $|k|$). Psychological literature has documented numerous such asymmetries, including loss aversion, negativity bias, and framing effects, all of which can manifest in preference annotations. The asymmetric model accommodates such patterns by learning all $2K$ thresholds independently from data, without imposing any relationship between $\zeta_{-k}$ and $\zeta_k$. This provides maximum flexibility to capture the actual structure of human preferences as reflected in the training data.

The choice between symmetric and asymmetric models represents a bias-variance trade-off: the symmetric model has fewer parameters and stronger inductive bias, while the asymmetric model has more flexibility to capture nuanced patterns in human annotations. In our experiments, we evaluate both variants and find that the symmetry assumption is reasonable in practice—the symmetric model often outperforms the asymmetric variant.

### 3.3.3 OPTIMIZATION VIA REPARAMETERIZATION

To handle the ordering constraints in (17), we reparameterize the thresholds using a monotonic transformation. For the asymmetric model, we parameterize

$$\zeta_{-K} = \alpha_0 \qquad \text{and} \qquad \zeta_k = \zeta_{k-1} + \exp(\alpha_k), \ \forall k \in [-K+1] \cup [K], \tag{19}$$

where $\alpha_0 \in \mathbb{R}$ and $\alpha_k \in \mathbb{R}$ for all $k \neq 0$. The exponential mapping ensures strict positivity of increments, maintaining $\zeta_k > \zeta_{k-1}$ by construction. For the symmetric model, we only parameterize the positive thresholds starting from $\zeta_1 = \exp(\alpha_1)$, with negative thresholds determined by $\zeta_{-k} = -\zeta_k$, reducing the number of parameters from $2K$ to $K$.

This reparameterization transforms the constrained optimization into an unconstrained problem over $(\phi, \alpha)$, solvable via standard gradient-based methods. In our implementation, we update reward and threshold parameters simultaneously at each iteration (see Appendix D for the detailed training algorithm). Alternative optimization strategies, including projected gradient descent and asynchronous updates, are discussed in Appendix E.

## 4 EXPERIMENTS

We evaluate our ordinal regression framework on real preference datasets with Likert-scale annotations, comparing against heuristic baselines across multiple model architectures and examining trade-offs between symmetric and asymmetric threshold formulations.

### 4.1 EXPERIMENTAL SETUP

**Base Models.** We evaluate our approach on three instruction-tuned language models: Llama-3.1-8B (Meta AI, 2024), Mistral-7B (Jiang et al., 2023), and Zephyr-7B (Tunstall et al., 2023). Each model uses its supervised fine-tuned checkpoint as initialization, with the language modeling head replaced by a single-layer reward head outputting scalar rewards.

**Datasets.** We train on HelpSteer2 (Wang et al., 2024c) and HelpSteer3 (Wang et al., 2025), which provide pairwise comparisons with 7-level preference strength labels from -3 (significantly worse) to +3 (significantly better). Both datasets contain fine-grained human annotations suitable for evaluating ordinal regression methods.

**Methods.** We evaluate three instantiations of our ordinal regression framework from Section 3: (1) **NLL-Symmetric**, using the negative log-likelihood loss (Eq. 15) with symmetric threshold constraints $\zeta_{-k} = -\zeta_k$; (2) **NLL-Asymmetric**, using the same NLL loss but learning all $2K$ thresholds independently; and (3) **All-Threshold**, using the margin-based all-threshold loss (Eq. 16) with asymmetric thresholds. We compare against three baseline heuristics: Margin BT, Scaled BT, and Soft Label. For ordinal methods, we jointly optimize reward and threshold parameters with L2 regularization on thresholds to ensure convergence (see Appendix G for empirical validation).

**Training and Evaluation.** All models are trained for 8 epochs with effective batch size 64 using Fully Sharded Data Parallel across 8 GPUs (H100 or H200). We evaluate on RewardBench (Lambert et al., 2024) and RM-Bench (Liu et al., 2024b). Comprehensive hyperparameter search details and infrastructure specifications are provided in Appendix F.

## 4.2 MAIN RESULTS

Tables 1 and 2 present our evaluation results. Across both benchmarks, the performance of our ordinal regression methods consistently match or exceed the baselines, with NLL-Symmetric emerging as our most effective approach. This method achieves the highest average scores in the majority of configurations, outperforming all baselines by 2-5% on average. The consistent superiority of NLL-Symmetric over NLL-Asymmetric validates our theoretical analysis in Theorem 3.2, suggesting that human preferences indeed exhibit the symmetry property assumed in our formulation.

The advantage of principled threshold learning becomes apparent when examining convergence behavior. As shown in Appendix G, our learned thresholds converge to stable values under appropriate regularization, with the choice of $\lambda$ offering a trade-off between convergence speed and flexibility. Without such regularization, thresholds exhibit unbounded growth as predicted by Theorem 3.1, highlighting the importance of our theoretical framework in ensuring stable optimization.

Beyond binary preference accuracy, our ordinal methods demonstrate genuine understanding of preference strength. Appendix H shows that NLL-Symmetric achieves approximately 55% exact accuracy and 85% accuracy within one ordinal level on validation data, confirming that our approach learns meaningful ordinal structure rather than merely preserving rankings. While All-Threshold generally performs below the NLL variants, it still outperforms baselines in several cases, indicating that both probabilistic and margin-based formulations improve upon ad-hoc heuristics. In the following section, we investigate the benefits of our ordinal framework beyond standard ranking accuracy, examining error severity, the necessity of joint training, and robustness to annotation noise.

## 4.3 ANALYSIS BEYOND BINARY ACCURACY

Standard binary accuracy treats all errors equally regardless of severity and all correct predictions equally regardless of calibration. We now examine three aspects of model quality that this metric misses.

**Error Severity.** When a reward model errs, the magnitude of the misranking matters: a model that is narrowly wrong on ambiguous cases is far preferable to one that confidently assigns high reward to dispreferred responses. We compare error margin distributions of standard Bradley-Terry (Simple BT) and NLL-Symmetric, both trained on HelpSteer2 with Llama-3.1-8B and evaluated on RewardBench. The ordinal approach reduces the number of errors by 35% (282 vs. 433), but the improvement in error *severity* is far more dramatic: the mean error margin drops from 3.827 to 0.501, a reduction of 87%. Simple BT produces errors with margins as large as 20, while NLL Ordinal Symmetric produces none exceeding 2.5. This means that when our model errs, it does so on genuinely ambiguous cases with low confidence, rather than making high-confidence misrankings—a critical property for downstream RL, where confidently incorrect rewards can severely misguide policy optimization. See Appendix I for the full distributional analysis.

Table 1: RM-Bench evaluation results. Best results *within each dataset×model group and column* are in **bold**, second best are underlined.

| Dataset | Model | Method | Chat | Math | Code | Safety | Hard | Normal | Easy | Total |
|---|---|---|---|---|---|---|---|---|---|---|
| HelpSteer2 | Llama | Margin BT | 0.662 | 0.563 | 0.521 | 0.885 | 0.363 | **0.701** | **0.908** | 0.657 |
| | | Scaled BT | 0.636 | 0.552 | 0.497 | 0.798 | 0.389 | 0.640 | 0.832 | 0.621 |
| | | Soft Label | 0.571 | 0.550 | 0.505 | 0.727 | 0.225 | 0.638 | 0.902 | 0.588 |
| | | NLL-Sym | 0.638 | 0.550 | 0.508 | **0.907** | **0.487** | 0.668 | 0.798 | 0.651 |
| | | NLL-Asym | 0.574 | 0.544 | **0.523** | 0.823 | 0.397 | 0.631 | 0.819 | 0.616 |
| | | All-Thresh | **0.688** | **0.566** | 0.519 | 0.860 | 0.421 | 0.681 | 0.874 | **0.658** |
| | Mistral | Margin BT | 0.634 | 0.539 | 0.496 | 0.760 | 0.317 | 0.643 | 0.862 | 0.607 |
| | | Scaled BT | 0.613 | **0.548** | 0.520 | 0.837 | **0.439** | 0.654 | 0.795 | 0.630 |
| | | Soft Label | 0.525 | 0.501 | 0.491 | 0.623 | 0.265 | 0.537 | 0.803 | 0.535 |
| | | NLL-Sym | **0.654** | 0.538 | **0.540** | **0.858** | 0.406 | **0.673** | **0.864** | **0.647** |
| | | NLL-Asym | 0.584 | 0.539 | 0.483 | 0.716 | 0.280 | 0.617 | 0.845 | 0.580 |
| | | All-Thresh | 0.619 | 0.519 | 0.493 | 0.775 | 0.315 | 0.630 | 0.859 | 0.601 |
| | Zephyr | Margin BT | 0.615 | 0.519 | 0.511 | **0.856** | 0.392 | 0.642 | 0.841 | 0.625 |
| | | Scaled BT | 0.621 | **0.534** | 0.501 | 0.762 | 0.327 | 0.637 | 0.850 | 0.605 |
| | | Soft Label | 0.635 | 0.531 | 0.504 | 0.840 | 0.349 | 0.658 | **0.875** | 0.627 |
| | | NLL-Sym | **0.661** | 0.533 | **0.519** | 0.835 | **0.399** | **0.667** | 0.846 | **0.637** |
| | | NLL-Asym | 0.604 | 0.530 | 0.509 | 0.650 | 0.230 | 0.617 | 0.873 | 0.573 |
| | | All-Thresh | 0.628 | **0.534** | 0.498 | 0.796 | 0.329 | 0.638 | **0.875** | 0.614 |
| HelpSteer3 | Llama | Margin BT | 0.717 | 0.587 | 0.532 | 0.765 | 0.390 | 0.681 | 0.880 | 0.650 |
| | | Scaled BT | **0.742** | **0.592** | 0.534 | 0.823 | 0.476 | 0.697 | 0.844 | 0.673 |
| | | Soft Label | 0.708 | 0.578 | 0.536 | 0.862 | 0.421 | 0.698 | **0.894** | 0.671 |
| | | NLL-Sym | 0.695 | 0.586 | 0.541 | **0.881** | 0.413 | **0.713** | 0.802 | **0.676** |
| | | NLL-Asym | 0.684 | 0.587 | **0.569** | 0.815 | **0.539** | 0.683 | 0.776 | 0.666 |
| | | All-Thresh | 0.663 | 0.573 | 0.540 | 0.815 | 0.526 | 0.663 | 0.755 | 0.648 |
| | Mistral | Margin BT | **0.700** | 0.555 | 0.506 | 0.718 | 0.414 | **0.636** | 0.809 | **0.620** |
| | | Scaled BT | 0.692 | 0.551 | 0.501 | 0.593 | 0.361 | 0.597 | 0.794 | 0.584 |
| | | Soft Label | 0.643 | 0.556 | 0.494 | 0.638 | **0.474** | 0.589 | 0.686 | 0.583 |
| | | NLL-Sym | 0.638 | 0.535 | 0.504 | **0.725** | 0.294 | 0.631 | **0.878** | 0.601 |
| | | NLL-Asym | 0.622 | **0.580** | 0.522 | 0.706 | 0.392 | 0.633 | 0.798 | 0.608 |
| | | All-Thresh | 0.621 | 0.560 | **0.523** | 0.555 | 0.255 | 0.599 | 0.841 | 0.565 |
| | Zephyr | Margin BT | 0.668 | 0.559 | **0.532** | 0.678 | **0.386** | 0.629 | 0.813 | 0.609 |
| | | Scaled BT | **0.694** | 0.551 | 0.492 | 0.693 | 0.355 | 0.631 | 0.837 | 0.607 |
| | | Soft Label | 0.653 | **0.564** | 0.525 | 0.605 | 0.318 | 0.618 | 0.825 | 0.587 |
| | | NLL-Sym | 0.684 | 0.548 | 0.502 | **0.727** | 0.357 | **0.640** | **0.847** | **0.615** |
| | | NLL-Asym | 0.643 | 0.554 | 0.336 | 0.519 | 0.303 | 0.569 | 0.817 | 0.563 |
| | | All-Thresh | 0.628 | 0.552 | 0.521 | 0.513 | 0.297 | 0.570 | 0.794 | 0.553 |

Table 2: RewardBench evaluation results for models trained on HelpSteer2 (left) and HelpSteer3 (right). Best results per metric are in **bold**, second best are underlined.

**HelpSteer2**

| Model | Method | Chat | Hard | Safety | Reason | Avg |
|---|---|---|---|---|---|---|
| Llama | Margin BT | **0.961** | 0.660 | 0.885 | 0.703 | 0.802 |
| | Scaled BT | 0.927 | 0.638 | 0.853 | 0.610 | 0.757 |
| | Soft Label | 0.939 | 0.581 | 0.689 | 0.680 | 0.722 |
| | NLL-Sym | 0.941 | **0.728** | **0.897** | **0.804** | **0.843** |
| | NLL-Asym | 0.911 | 0.695 | 0.837 | 0.794 | 0.809 |
| | All-Thresh | 0.922 | 0.689 | 0.872 | 0.798 | 0.820 |
| Mistral | Margin BT | 0.933 | 0.601 | 0.738 | 0.725 | 0.749 |
| | Scaled BT | 0.930 | 0.660 | **0.824** | 0.787 | **0.800** |
| | Soft Label | 0.905 | 0.626 | 0.569 | 0.776 | 0.694 |
| | NLL-Sym | **0.939** | **0.684** | 0.708 | 0.783 | 0.779 |
| | NLL-Asym | 0.919 | 0.629 | 0.819 | 0.699 | 0.767 |
| | All-Thresh | 0.897 | 0.680 | 0.730 | **0.827** | 0.783 |
| Zephyr | Margin BT | 0.922 | **0.700** | **0.857** | 0.593 | 0.768 |
| | Scaled BT | 0.944 | 0.665 | 0.727 | 0.786 | 0.781 |
| | Soft Label | 0.933 | 0.660 | 0.819 | 0.816 | **0.807** |
| | NLL-Sym | **0.953** | 0.603 | 0.761 | 0.756 | 0.768 |
| | NLL-Asym | 0.911 | 0.662 | 0.680 | **0.856** | 0.777 |
| | All-Thresh | 0.933 | 0.662 | 0.751 | 0.821 | 0.792 |

**HelpSteer3**

| Model | Method | Chat | Hard | Safety | Reason | Avg |
|---|---|---|---|---|---|---|
| Llama | Margin BT | **0.947** | 0.700 | 0.780 | 0.656 | 0.771 |
| | Scaled BT | 0.944 | 0.748 | 0.793 | 0.638 | 0.781 |
| | Soft Label | 0.922 | 0.728 | **0.823** | **0.747** | 0.805 |
| | NLL-Sym | **0.947** | **0.765** | 0.808 | 0.707 | **0.807** |
| | NLL-Asym | 0.936 | 0.697 | 0.785 | 0.641 | 0.765 |
| | All-Thresh | 0.872 | 0.728 | 0.780 | 0.676 | 0.764 |
| Mistral | Margin BT | 0.916 | 0.702 | **0.700** | 0.588 | 0.727 |
| | Scaled BT | 0.908 | **0.724** | 0.687 | **0.704** | **0.756** |
| | Soft Label | 0.791 | 0.700 | 0.653 | 0.654 | 0.699 |
| | NLL-Sym | **0.927** | 0.706 | 0.695 | 0.668 | 0.749 |
| | NLL-Asym | 0.888 | 0.627 | 0.685 | 0.616 | 0.704 |
| | All-Thresh | 0.925 | 0.568 | 0.603 | 0.651 | 0.687 |
| Zephyr | Margin BT | 0.911 | 0.660 | 0.643 | 0.612 | 0.707 |
| | Scaled BT | 0.888 | 0.634 | 0.723 | **0.673** | 0.730 |
| | Soft Label | 0.908 | 0.686 | 0.642 | 0.566 | 0.700 |
| | NLL-Sym | 0.913 | **0.722** | **0.753** | 0.668 | **0.764** |
| | NLL-Asym | 0.916 | 0.640 | 0.574 | 0.611 | 0.685 |
| | All-Thresh | **0.925** | 0.600 | 0.557 | 0.615 | 0.674 |

Table 3: Ordinal prediction on HelpSteer2 test set (448 examples). Joint training substantially outperforms post-hoc calibration of baseline reward models across all metrics.

| Method | MAE $\downarrow$ | Acc@0 $\uparrow$ | Acc@1 $\uparrow$ | Acc@2 $\uparrow$ |
|---|---|---|---|---|
| Scaled BT (post-hoc) | 2.667 | 10.9% | 37.1% | 44.0% |
| Margin BT (post-hoc) | 2.181 | 16.1% | 48.7% | 59.6% |
| Soft Label (post-hoc) | 1.725 | 12.9% | 47.8% | 78.1% |
| NLL-Symmetric (joint) | **1.060** | **29.7%** | **72.5%** | **92.9%** |

**Joint Training vs. Post-Hoc Calibration.** A natural question is whether our framework's benefits could be replicated by simply fitting thresholds to a standard reward model after training. We train baseline models (Margin BT, Scaled BT, Soft Label) on HelpSteer2 with Llama-3.1-8B, then perform post-hoc calibration by learning thresholds on frozen reward differences using the same ordinal NLL objective. Table 3 compares these against jointly trained NLL-Symmetric on the held-out test set. Joint training achieves 38% lower MAE (1.060 vs. 1.725) and nearly doubles exact ordinal accuracy (29.7% vs. 16.1%). The Acc@1 gap is especially notable: our method predicts preference strength within one level for 72.5% of examples, versus fewer than half for the best baseline. Post-hoc calibration cannot recover the fine-grained structure that joint optimization discovers. See Appendix H for the full procedure and training-set results.

**Robustness to Label Noise.** We evaluate robustness under two noise models at corruption rates from 25% to 100%: *systematic shift noise*, where labels are perturbed by one ordinal level (modeling calibration errors), and *random label noise*, where labels are replaced uniformly at random (modeling catastrophic failures). Under systematic shift noise, performance is essentially unchanged even at 100% corruption (RewardBench average: 0.808–0.846 vs. 0.843 baseline), as the learned thresholds absorb systematic biases. Under random noise, degradation is graceful: at 25% corruption performance matches the clean baseline, and meaningful learning persists through 50% corruption. Full results including per-category breakdowns and training dynamics are in Appendix J.

## 5 CONCLUSIONS

We presented the first principled framework for reward modeling with ordinal preference data, moving beyond ad-hoc modifications of binary preference models. By formulating the problem as discrete ordinal regression, we derived theoretically grounded loss functions that learn threshold parameters directly from data, eliminating the need for manually specified margins or scaling factors. Our theoretical analysis revealed fundamental properties of the optimization landscape, including the necessity of regularization for bounded solutions and a proof that symmetric thresholds arise naturally when human preferences exhibit symmetry. Empirical results across multiple benchmarks and model architectures demonstrate that our approach consistently achieves competitive or superior performance compared to heuristic baselines, with the symmetric NLL formulation proving particularly effective—validating both our theoretical framework and suggesting that human preferences indeed exhibit the symmetry property. Beyond ranking accuracy, our ordinal framework yields substantially better-calibrated models, reducing mean error severity by 87% compared to standard Bradley-Terry, and these benefits cannot be replicated by post-hoc threshold fitting. The framework also exhibits strong robustness to realistic annotation noise. Future work could extend this framework to DPO as outlined in Appendix A, handle more complex preference structures such as multi-aspect ratings or pairwise comparisons with uncertainty estimates, and adapt to evolving data collection methodologies. As the field progresses toward more sophisticated annotation schemes that capture nuanced human feedback beyond binary preferences, principled methods for leveraging these richer signals will become increasingly critical for effective language model alignment.

**Reproducibility Statement.** We have made extensive efforts to ensure the reproducibility of our work. Complete proofs for all theoretical results (Theorems 3.1 and 3.2) are provided in Appendices B and C, with clear statements of assumptions and detailed derivations. The full training algorithm for our ordinal regression approach is detailed in Appendix D, with alternative optimization strategies discussed in Appendix E. Comprehensive experimental details are provided in Appendix F, including: complete model and data configurations (Section F.1), exhaustive hyperparameter search spaces and final selections (Section F.3), and infrastructure specifications including GPU requirements and implementation details (Section G.1). All experiments use fixed random seeds and consistent preprocessing pipelines as described in Appendix F. Data processing steps, including tokenization limits, padding strategies, and prompt truncation procedures, are fully specified in Section F.1. The convergence behavior of our learned thresholds is empirically validated in Appendix G, and additional ordinal performance metrics are reported in Appendix H. Error margin analysis and noise robustness experiments are detailed in Appendices I and J, respectively. Our implementation builds on the open-source TRL library with extensions for ordinal regression methods. As stated in Section G.1, all model checkpoints, configuration files, and code will be released upon publication to facilitate reproduction and extension of our work.

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

## A  EXTENSION TO DIRECT POLICY OPTIMIZATION

### A.1  DIRECT POLICY OPTIMIZATION PRELIMINARIES

Unlike the standard RLHF approach that requires training a reward model followed by RL-based policy optimization, Direct Policy Optimization (DPO) (Rafailov et al., 2023) directly optimizes the policy without explicitly learning a reward model. Despite this conceptual difference, DPO relies on the same Bradley-Terry preference model and uses the same type of preference dataset as reward modeling.

Given a training set $\mathcal{D} = \{x_i, y_{w,i}, y_{l,i}\}_{i=1}^n$ with preferred and dispreferred responses, DPO minimizes the following loss as a function of a parameterized policy $\pi_\theta$,

$$\mathcal{L}_{\text{DPO}}(\pi_\theta; \mathcal{D}) = -\mathbb{E}_{(x,y_w,y_l)\sim\mathcal{D}} \left[ \log \sigma \left( \beta \log \left( \frac{\pi_\theta(y_w|x)}{\pi_{\text{ref}}(y_w|x)} \right) - \beta \log \left( \frac{\pi_\theta(y_l|x)}{\pi_{\text{ref}}(y_l|x)} \right) \right) \right], \quad (20)$$

where $\pi_{\text{ref}}$ is a reference policy (typically the SFT checkpoint) and $\beta$ is a parameter that controls the deviation from $\pi_{\text{ref}}$.

Following the convention in Rafailov et al. (2023), we can define the pseudo-reward of DPO as

$$\hat{r}_\theta(x, y) = \beta \log \left( \frac{\pi_\theta(y|x)}{\pi_{\text{ref}}(y|x)} \right). \quad (21)$$

With this definition, we note that both the reward modeling loss (9) and the DPO loss (20) take the same form of log-sigmoid of reward differences, with parameterized rewards $r_\phi$ and $\hat{r}_\theta$, respectively. This structural similarity implies that the ordinal regression framework we develop for reward modeling in the main paper can be naturally extended to DPO by replacing $r_\phi(x, y)$ with $\hat{r}_\theta(x, y)$.

### A.2  ORDINAL DPO

Given the structural similarity between reward modeling and DPO, we can extend the ordinal regression framework from Section 3 to direct policy optimization. For DPO with ordinal feedback, the predictor value becomes the pseudo-reward difference,

$$\hat{s}_\theta(x, y, y') = \hat{r}_\theta(x, y) - \hat{r}_\theta(x, y') = \beta \log \left( \frac{\pi_\theta(y|x)/\pi_{\text{ref}}(y|x)}{\pi_\theta(y'|x)/\pi_{\text{ref}}(y'|x)} \right). \quad (22)$$

Using the same threshold structure with $2K$ thresholds $\zeta_{-K}, \ldots, \zeta_{-1}, \zeta_1, \ldots, \zeta_K$, we can adapt both loss functions from Section 3.3.1.

**Probabilistic Approach for DPO.**  The NLL loss for ordinal DPO becomes

$$\mathcal{L}_{\text{NLL-DPO}}(\pi_\theta, \zeta; x, y, y', z) = -\log p(y \succ_z y' \mid x), \quad (23)$$

where the probability is computed using $\hat{s}_\theta(x, y, y')$ in place of $s_\phi(x, y, y')$ in equation (14).

**Margin-Based Approach for DPO.**  The all-threshold loss for ordinal DPO is

$$\mathcal{L}_{\text{AT-DPO}}(\pi_\theta, \zeta; x, y, y', z) = \sum_{l\in\mathbb{K}\setminus\{0\}} -\log \sigma\big(\nu(l; z) \cdot (\zeta_l - \hat{s}_\theta(x, y, y'))\big), \quad (24)$$

where $\nu(l; z)$ is defined as in the reward modeling case.

The optimization problem for ordinal DPO becomes

$$\min_{\theta, \zeta\in\mathcal{C}} \sum_{(x,y,y',z)\in\mathcal{D}} \mathcal{L}(\pi_\theta, \zeta; x, y, y', z) + \lambda\|\zeta\|_2^2, \quad (25)$$

where the regularization term on thresholds is necessary for the same reasons discussed in Theorem 3.1.

**Symmetric and Asymmetric Models.** The choice between symmetric and asymmetric models applies equally to DPO. Under the symmetric model, we impose $\zeta_{-k} = -\zeta_k$, while the asymmetric model learns all thresholds independently. The same reparameterization strategy from Section 3.3.3 can be used to handle the ordering constraints.

**Implementation Considerations.** The key difference in implementing ordinal DPO compared to ordinal reward modeling lies in computing the pseudo-rewards. At each optimization step, we must:

1. Compute the policy probabilities $\pi_\theta(y|x)$ and $\pi_\theta(y'|x)$
2. Compute the reference probabilities $\pi_{\text{ref}}(y|x)$ and $\pi_{\text{ref}}(y'|x)$
3. Calculate the pseudo-reward difference $\hat{s}_\theta(x, y, y')$
4. Apply the chosen ordinal loss function

The gradient computation involves backpropagating through both the policy network and the threshold parameters, similar to the reward modeling case but with the additional complexity of the log-ratio terms.

### A.3 THEORETICAL CONSIDERATIONS

While we leave empirical evaluation of ordinal DPO for future work, we note that the theoretical properties established for ordinal reward modeling largely transfer to the DPO setting. In particular:

1. The unbounded solution set problem (Theorem 3.1) applies equally to DPO, necessitating threshold regularization.
2. The symmetry theorem (Theorem 3.2) can be adapted to DPO under appropriate assumptions about the symmetry of human preferences in the policy optimization context.
3. The convergence guarantees of standard DPO (Rafailov et al., 2023) extend to the ordinal case when the thresholds are properly regularized.

## B PROOF OF THEOREM 3.1

We prove Theorem 3.1 by analyzing how the losses behave under scaling. Consider a solution $(r_\phi^*, \zeta^*)$ and its scaled version $(cr_\phi^*, c\zeta^*)$ for $c > 0$.

**All-Threshold Loss.** For a training example $(x, y, y', z)$ with reward difference $s = r_\phi(x, y) - r_\phi(x, y')$, the all-threshold loss under scaling becomes

$$\mathcal{L}_{\text{AT}}(cs, c\zeta) = \sum_{l \in \mathbb{K} \setminus \{0\}} - \log \sigma\big(c \cdot \nu(l; z) \cdot (\zeta_l - s)\big). \tag{26}$$

Consider each term in the sum. If $\nu(l; z) \cdot (\zeta_l - s) > 0$ (the threshold is correctly satisfied), then as $c \to \infty$, we have $\sigma(c \cdot \nu(l; z) \cdot (\zeta_l - s)) \to 1$, so the term contributes $-\log(1) = 0$ to the loss.

If $\nu(l; z) \cdot (\zeta_l - s) < 0$ (threshold violation), then the argument to the sigmoid is large and negative. Using the approximation $\sigma(u) \approx e^u$ for large negative $u$, we obtain

$$- \log \sigma\big(c \cdot \nu(l; z) \cdot (\zeta_l - s)\big) \approx - \log(e^{c \cdot \nu(l;z) \cdot (\zeta_l - s)}) \tag{27}$$

$$= -c \cdot \nu(l; z) \cdot (\zeta_l - s) \tag{28}$$

$$= c|\zeta_l - s|, \tag{29}$$

which grows linearly with $c$.

For correct classification with $z > 0$, we need $s \in (\zeta_{z-1}, \zeta_z)$. This ensures that $s > \zeta_l$ for all $l < z$ (making $\nu(l; z) \cdot (\zeta_l - s) > 0$) and $s < \zeta_l$ for all $l \geq z$ (also making $\nu(l; z) \cdot (\zeta_l - s) > 0$). Therefore, all terms in the sum approach zero as $c \to \infty$. The case $z < 0$ is analogous, and for $z = 0$, correct classification means $s \in (\zeta_{-1}, \zeta_1)$, which similarly ensures all terms vanish.

**Negative Log-Likelihood Loss.**   For the NLL loss with $z > 0$ (other cases are analogous),

$$\mathcal{L}_{\text{NLL}}(cs, c\zeta) = -\log[\sigma(c(\zeta_z - s)) - \sigma(c(\zeta_{z-1} - s))]. \tag{30}$$

If $\zeta_{z-1} < s < \zeta_z$ (correct classification), then as $c \to \infty$, we have $\zeta_z - s > 0$, which implies $\sigma(c(\zeta_z - s)) \to 1$, and $\zeta_{z-1} - s < 0$, which implies $\sigma(c(\zeta_{z-1} - s)) \to 0$. Therefore, $\mathcal{L}_{\text{NLL}}(cs, c\zeta) \to -\log(1 - 0) = 0$.

If $s < \zeta_{z-1}$ (misclassification), both $(\zeta_z - s)$ and $(\zeta_{z-1} - s)$ are positive, so both sigmoids approach 1. Using the approximation $\sigma(u) \approx 1 - e^{-u}$ for large positive $u$, we obtain

$$\sigma(c(\zeta_z - s)) - \sigma(c(\zeta_{z-1} - s)) \approx [1 - e^{-c(\zeta_z - s)}] - [1 - e^{-c(\zeta_{z-1} - s)}] \tag{31}$$

$$= e^{-c(\zeta_{z-1} - s)} - e^{-c(\zeta_z - s)} \tag{32}$$

$$\approx e^{-c(\zeta_{z-1} - s)}, \tag{33}$$

where the last approximation holds because $e^{-c(\zeta_z - s)} \ll e^{-c(\zeta_{z-1} - s)}$ for large $c$. Thus,

$$\mathcal{L}_{\text{NLL}}(cs, c\zeta) \approx -\log(e^{-c(\zeta_{z-1} - s)}) = c(\zeta_{z-1} - s), \tag{34}$$

which grows linearly with $c$.

Similarly, if $s > \zeta_z$ (misclassification), both $(\zeta_z - s)$ and $(\zeta_{z-1} - s)$ are negative, so both sigmoids approach 0. Using the approximation $\sigma(u) \approx e^u$ for large negative $u$, the difference becomes approximately $e^{c(\zeta_z - s)}$, leading to $\mathcal{L}_{\text{NLL}}(cs, c\zeta) \approx c(s - \zeta_z)$, which again grows linearly with $c$.

## C   PROOF OF THEOREM 3.2

We prove that symmetric preference data implies symmetric thresholds under the ordered logit model.

Consider the ordered logit model where the probability of observing preference level $k > 0$ given reward difference $r = s_\phi(x, y, y')$ is

$$p_k(r) = \sigma(\zeta_k - r) - \sigma(\zeta_{k-1} - r). \tag{35}$$

Similarly, for preference level $-k < 0$:

$$p_{-k}(r) = \sigma(\zeta_{-k+1} - r) - \sigma(\zeta_{-k} - r). \tag{36}$$

The data symmetry assumption requires $p_k(r) = p_{-k}(-r)$ for all $r \in \mathbb{R}$ and all $k \in \{1, \dots, K\}$.

We begin with the boundary case $k = K$. Since $\zeta_K = \infty$, we have

$$p_K(r) = \sigma(\infty) - \sigma(\zeta_{K-1} - r) = 1 - \sigma(\zeta_{K-1} - r). \tag{37}$$

Similarly, since $\zeta_{-(K+1)} = \zeta_{-K-1} = -\infty$, we have

$$p_{-K}(r) = \sigma(\zeta_{-K+1} - r) - \sigma(-\infty) = \sigma(\zeta_{-K+1} - r). \tag{38}$$

The symmetry condition $p_K(r) = p_{-K}(-r)$ gives us

$$1 - \sigma(\zeta_{K-1} - r) = \sigma(\zeta_{-K+1} + r). \tag{39}$$

Using $\sigma(-u) = 1 - \sigma(u)$, we can rewrite the left side as $\sigma(-\zeta_{K-1} + r)$. Therefore,

$$\sigma(-\zeta_{K-1} + r) = \sigma(\zeta_{-K+1} + r). \tag{40}$$

Since the sigmoid function is strictly monotonic, this equality for all $r$ implies

$$-\zeta_{K-1} = \zeta_{-K+1} \quad \Rightarrow \quad \zeta_{-K+1} = -\zeta_{K-1}. \tag{41}$$

Now, for $k = K - 1$, using the symmetry condition and the fact that we've established $\zeta_{-K+1} = -\zeta_{K-1}$:

$$p_{K-1}(r) = \sigma(\zeta_{K-1} - r) - \sigma(\zeta_{K-2} - r), \tag{42}$$

$$p_{-(K-1)}(-r) = \sigma(\zeta_{-K+2} + r) - \sigma(\zeta_{-K+1} + r) \tag{43}$$

$$= \sigma(\zeta_{-K+2} + r) - \sigma(-\zeta_{K-1} + r). \tag{44}$$

Using $\sigma(-u) = 1 - \sigma(u)$ on the second term:

$$p_{-(K-1)}(-r) = \sigma(\zeta_{-K+2} + r) - [1 - \sigma(\zeta_{K-1} - r)]. \tag{45}$$

Setting $p_{K-1}(r) = p_{-(K-1)}(-r)$ and simplifying:

$$\sigma(\zeta_{K-1} - r) - \sigma(\zeta_{K-2} - r) = \sigma(\zeta_{-K+2} + r) + \sigma(\zeta_{K-1} - r) - 1. \tag{46}$$

This reduces to

$$1 - \sigma(\zeta_{K-2} - r) = \sigma(\zeta_{-K+2} + r), \tag{47}$$

which, by the same argument as before, implies $\zeta_{-K+2} = -\zeta_{K-2}$.

Proceeding inductively, we can show that $\zeta_{-k} = -\zeta_k$ for all $k \in \{1, \ldots, K\}$. The case $k = 0$ is automatically satisfied since we excluded $\zeta_0$ from our parameterization, with the interval $(\zeta_{-1}, \zeta_1)$ corresponding to preference level $z = 0$.

## D  TRAINING ALGORITHM

Algorithm 1 provides the complete training procedure for our ordinal regression approach to reward modeling. The algorithm jointly optimizes the reward model parameters $\phi$ and threshold parameters $\alpha$ using gradient descent, with the thresholds reparameterized through exponential mappings to maintain ordering constraints. The symmetric variant reduces the number of threshold parameters by enforcing $\zeta_{-k} = -\zeta_k$, while the asymmetric variant learns all thresholds independently.

## E  ALTERNATIVE OPTIMIZATION STRATEGIES

While we use unconstrained optimization via reparameterization in our main experiments, we explored several alternative approaches for handling the threshold ordering constraints.

### E.1  PROJECTED GRADIENT DESCENT

An alternative to reparameterization is to directly optimize the thresholds while projecting them onto the constraint set at each iteration. Since the constraint set $\mathcal{C} = \{\zeta : \zeta_{-K} < \cdots < \zeta_{-1} < \zeta_1 < \cdots < \zeta_K\}$ is open, we cannot directly project onto it. Instead, we define a closed approximation,

$$\mathcal{C}_\epsilon = \{\zeta \in \mathbb{R}^{2K} : \zeta_{k+1} \geq \zeta_k + \epsilon \text{ for consecutive } k\}, \tag{48}$$

where $\epsilon > 0$ is a small constant ensuring minimum separation between thresholds.

The projected gradient descent algorithm proceeds as follows:

---

**Algorithm 2** Projected Gradient Descent for Ordinal Regression

---

**Require:** Learning rate $\eta$, minimum gap $\epsilon$, initial parameters $\phi_0, \zeta_0 \in \mathcal{C}_\epsilon$
**Ensure:** Optimized parameters $\phi^*, \zeta^*$
 1: Initialize $\zeta_0$ with uniform spacing in $\mathcal{C}_\epsilon$
 2: **for** $t = 0, 1, 2, \ldots$ **do**
 3:     Compute gradients $\nabla_\phi \mathcal{L}$ and $\nabla_\zeta \mathcal{L}$
 4:     Update $\phi_{t+1} = \phi_t - \eta \nabla_\phi \mathcal{L}$
 5:     Update $\tilde{\zeta}_{t+1} = \zeta_t - \eta \nabla_\zeta \mathcal{L}$
 6:     Project $\zeta_{t+1} = \text{Proj}_{\mathcal{C}_\epsilon}(\tilde{\zeta}_{t+1})$
 7: **end for**

---

---

**Algorithm 1** Ordinal Regression for Reward Modeling

---

**Require:** Dataset $\mathcal{D} = \{(x_i, y_i, y_i', z_i)\}_{i=1}^n$, loss type $\mathcal{L} \in \{\mathcal{L}_{\text{NLL}}, \mathcal{L}_{\text{AT}}\}$, symmetry flag
**Require:** Learning rates $\eta_\phi, \eta_\alpha$, regularization weight $\lambda$, epochs $E$
**Ensure:** Trained reward model $r_\phi$ and thresholds $\zeta$
 1: Initialize reward model parameters $\phi$ from SFT checkpoint
 2: **if** symmetric **then**
 3:     Initialize threshold parameters $\alpha = [\alpha_1, \ldots, \alpha_K] \in \mathbb{R}^K$
 4: **else**
 5:     Initialize threshold parameters $\alpha = [\alpha_0, \alpha_1, \ldots, \alpha_{2K-1}] \in \mathbb{R}^{2K}$
 6: **end if**
 7: Initialize $\text{Opt}_\phi$ with cosine schedule, $\text{Opt}_\alpha$ with constant schedule
 8: **for** epoch $= 1$ to $E$ **do**
 9:     **for** batch $(x, y, y', z)$ in $\mathcal{D}$ **do**
10:         Compute reward difference: $s_\phi(x, y, y') = r_\phi(x, y) - r_\phi(x, y')$
11:         **if** symmetric **then**
12:             $\zeta_1 = \exp(\alpha_1)$
13:             **for** $k = 2$ to $K$ **do**
14:                 $\zeta_k = \zeta_{k-1} + \exp(\alpha_k)$
15:             **end for**
16:             **for** $k = 1$ to $K$ **do**
17:                 $\zeta_{-k} = -\zeta_k$
18:             **end for**
19:         **else**
20:             $\zeta_{-K} = \alpha_0$
21:             **for** $j = -K + 1$ to $-1$ **do**
22:                 $\zeta_j = \zeta_{j-1} + \exp(\alpha_{j+K})$
23:             **end for**
24:             $\zeta_1 = \zeta_{-1} + \exp(\alpha_K)$
25:             **for** $j = 2$ to $K$ **do**
26:                 $\zeta_j = \zeta_{j-1} + \exp(\alpha_{j+K-1})$
27:             **end for**
28:         **end if**
29:         Compute loss: $L = \mathcal{L}(r_\phi, \zeta; x, y, y', z) + \lambda \|\zeta\|_2^2$
30:         Update model: $\phi \leftarrow \phi - \eta_\phi \cdot \text{Opt}_\phi.\text{step}(\nabla_\phi L)$
31:         Update thresholds: $\alpha \leftarrow \alpha - \eta_\alpha \cdot \text{Opt}_\alpha.\text{step}(\nabla_\alpha L)$
32:     **end for**
33: **end for**

---

The projection onto $\mathcal{C}_\epsilon$ can be formulated as a quadratic program:

$$\text{Proj}_{\mathcal{C}_\epsilon}(\tilde{\zeta}) = \underset{\zeta \in \mathcal{C}_\epsilon}{\arg\min} \|\zeta - \tilde{\zeta}\|_2^2, \tag{49}$$

which can be solved efficiently using specialized QP solvers, particularly when $K$ is small (as is typical in practice).

### E.2 ASYNCHRONOUS PARAMETER UPDATES

During experimentation, we observed that updating reward parameters $\phi$ and threshold parameters $\zeta$ at different frequencies can improve optimization stability. This approach is motivated by the observation that thresholds typically converge more slowly than reward parameters and may benefit from less frequent but more stable updates.

The asynchronous update scheme works as follows:

---

**Algorithm 3** Asynchronous Parameter Updates

---

**Require:** Update frequency $N$ for thresholds, learning rates $\eta_\phi, \eta_\alpha$
1: **for** $t = 0, 1, 2, \ldots$ **do**
2:     Compute gradient $\nabla_\phi \mathcal{L}$ and update $\phi_{t+1} = \phi_t - \eta_\phi \nabla_\phi \mathcal{L}$
3:     **if** $t \bmod N = 0$ **then**
4:         Compute gradient $\nabla_\alpha \mathcal{L}$ using accumulated statistics
5:         Update $\alpha_{t+1} = \alpha_t - \eta_\alpha \nabla_\alpha \mathcal{L}$
6:     **else**
7:         Keep $\alpha_{t+1} = \alpha_t$
8:     **end if**
9: **end for**

---

In our experiments, we tested $N \in \{10, 50, 100\}$ and found that while this approach can stabilize training in some scenarios, particularly when the reward model is prone to large gradient updates, the simultaneous update strategy (equivalent to $N = 1$) generally performed comparably or better with appropriate learning rate scheduling. The asynchronous approach may be beneficial when:

1. The reward model architecture is particularly deep or unstable
2. The dataset contains significant label noise
3. Computational resources require less frequent threshold updates

### E.3 COMPARISON OF OPTIMIZATION STRATEGIES

Each optimization strategy has distinct trade-offs:

**Reparameterization (our choice):** Transforms the problem into unconstrained optimization, enabling the use of standard optimizers without modification. The exponential mapping naturally maintains constraints but can occasionally lead to numerical issues if $\alpha$ values become very negative (causing near-zero increments).

**Projected Gradient Descent:** Maintains thresholds in their natural space, making interpretation easier. However, it requires solving a projection problem at each iteration and choosing an appropriate $\epsilon$ value.

**Asynchronous Updates:** Can improve stability but requires tuning the update frequency $N$ and may slow convergence if thresholds are updated too infrequently.

In our experiments, we found reparameterization to be the most robust across different datasets and model architectures, which motivated our choice for the main implementation.

## F DETAILED EXPERIMENTAL SETUP

### F.1 MODEL AND DATA CONFIGURATION

**Base Models.** We conduct experiments using three pre-trained language models obtained from HuggingFace Hub: `allenai/Llama-3.1-Tulu-3-8B-SFT` (Llama-3.1-8B), `HuggingFaceH4/mistral-7b-sft-beta` (Mistral-7B), and `alignment-handbook/zephyr-7b-sft-full` (Zephyr-7B). Each model initializes from its supervised fine-tuned checkpoint. The language modeling head is replaced with a single-layer reward head that outputs scalar rewards. All experiments employ mixed precision training with bfloat16 for computational efficiency.

**Data Processing.** For both HelpSteer2 and HelpSteer3 datasets, we apply consistent preprocessing. Sequences are limited to 2048 tokens with prompts truncated to 1600 tokens when necessary. Tokenization uses model-specific tokenizers with padding tokens set to EOS when undefined. During training, prompt tokens are masked with -100 to exclude them from loss computation. Each response is ensured to terminate with exactly one EOS token, with any embedded EOS tokens removed during preprocessing.

## F.2    Hyperparameter Search

We conducted comprehensive hyperparameter optimization to identify robust training configurations. The search space and final selections are detailed below.

**Learning Rates and Scheduling.**    For model parameters, we evaluated learning rates in $\{5 \times 10^{-7}, 1 \times 10^{-6}, 2 \times 10^{-6}, 5 \times 10^{-6}\}$ with both cosine and linear schedulers. The optimal configuration uses $1 \times 10^{-6}$ with cosine scheduling and 10% warmup steps. For ordinal threshold parameters, we tested rates in $\{1 \times 10^{-4}, 5 \times 10^{-4}, 1 \times 10^{-3}, 5 \times 10^{-3}, 1 \times 10^{-2}\}$, finding $1 \times 10^{-3}$ with constant scheduling to work best.

**Regularization Strategies.**    We explored L2 regularization weights $\lambda \in \{0, 0.1, 1, 2, 5, 10\}$ for threshold parameters in ordinal methods. Lower values ($\lambda < 0.1$) led to optimization instabilities as predicted by Theorem 3.1, while very high values ($\lambda > 10$) overly constrained threshold learning. The optimal range was $\lambda \in [0.1, 5]$, with specific values varying by model and dataset. We also tested weight decay on model parameters but found no improvement over unregularized training.

**Training Dynamics.**    We investigated updating threshold parameters at different frequencies relative to model updates, testing intervals of $\{1, 5, 10, 15\}$ steps. With properly tuned hyperparameters, synchronous updates (interval=1) achieved performance matching that of asynchronous schemes while being simpler to implement. Details of the asynchronous update strategy are provided in Appendix E. Batch size experiments ranged from 32 to 128 (effective batch size after gradient accumulation), with 64 providing the best stability-efficiency trade-off. All models are trained for 8 epochs, which we determined sufficient for convergence through preliminary experiments.

## F.3    Infrastructure and Implementation

**Computational Resources.**    All experiments utilize 8 GPUs (either NVIDIA H100 with 80GB HBM3 or H200 with 141GB HBM3e) with Fully Sharded Data Parallel (FSDP) for distributed training. We employ automatic transformer layer wrapping with full sharding strategy to minimize memory footprint while maintaining training efficiency. Gradient accumulation over 4 steps achieves an effective batch size of 64 while fitting within GPU memory constraints.

**Model Selection.**    We select the best checkpoint based on reward accuracy on the validation set, where reward accuracy measures the frequency of assigning higher rewards to preferred responses in preference pairs. To ensure model quality, we exclude checkpoints with anomalously high validation loss (typically occurring in very early training) and, for ordinal methods, those from training transition phases where thresholds undergo rapid adjustments (see Appendix G for detailed analysis of threshold convergence behavior).

**Reproducibility.**    All experiments use fixed random seed 0 for weight initialization, data shuffling, and dropout. Training configurations are managed through Hydra framework with all hyperparameters logged to Weights & Biases for full reproducibility. Model checkpoints and configuration files will be released upon publication. Our implementation builds on the Transformer Reinforcement Learning (TRL) library with extensions for ordinal regression methods.

## G    Threshold Convergence Analysis

### G.1    Effect of Regularization on Threshold Stability

Figure 1 illustrates the evolution of threshold parameters during training for NLL-Symmetric on Llama with HelpSteer3, comparing three L2 regularization weights: $\lambda \in \{0, 0.1, 1\}$. The plots reveal distinct convergence behaviors that represent different trade-offs between stability and flexibility.

Without regularization ($\lambda = 0$, red curves), the thresholds exhibit continuous growth throughout training, with outer thresholds (zeta_0 and zeta_5, corresponding to $\zeta_{-3}$ and $\zeta_3$) showing the most dramatic increases. This unbounded behavior empirically validates Theorem 3.1, which

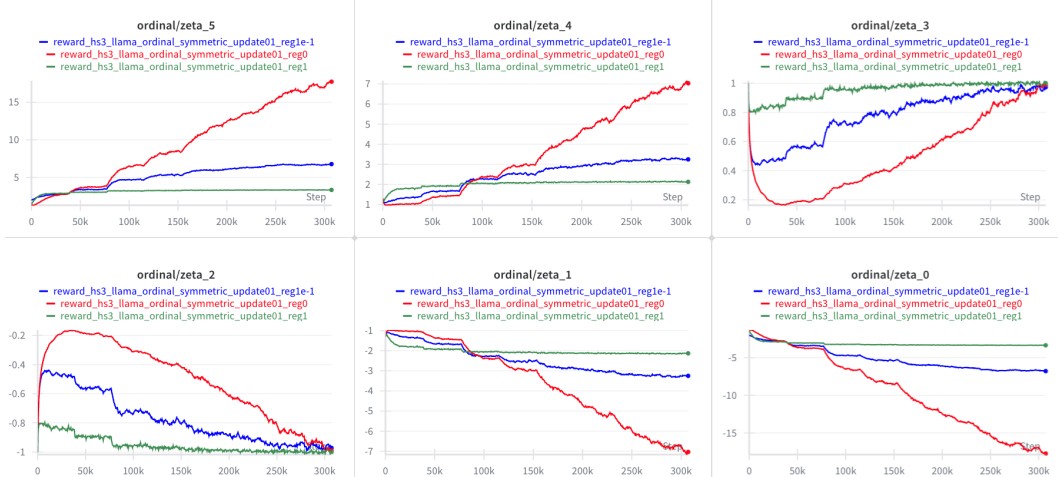

Figure 1: Evolution of symmetric threshold parameters during training with different L2 regularization weights. The plots show `zeta_0` through `zeta_5` which correspond to the ordinal thresholds $\zeta_{-3}$ through $\zeta_3$ in our formulation (excluding $\zeta_0$ which is not parameterized). Due to symmetry constraints, `zeta_0` $= -\zeta_3$, `zeta_1` $= -\zeta_2$, and `zeta_2` $= -\zeta_1$ in the plots. Without regularization ($\lambda = 0$, red), thresholds exhibit unbounded growth. Moderate regularization ($\lambda = 0.1$, blue) allows gradual convergence with more flexibility, while stronger regularization ($\lambda = 1$, green) enforces rapid convergence to stable values.

proves that the unregularized optimization problem may admit no finite optimal solution. The model achieves arbitrarily low loss by scaling both rewards and thresholds simultaneously, leading to numerical instability despite potentially maintaining correct relative orderings.

With moderate regularization ($\lambda = 0.1$, blue curves), thresholds ultimately converge but maintain more flexibility during training, allowing the model to explore a wider range of threshold configurations before settling into stable values. This gradual convergence can be beneficial when the optimal threshold structure is not immediately apparent from the data. In contrast, stronger regularization ($\lambda = 1$, green curves) enforces rapid convergence within the first 100k steps, providing greater stability at the potential cost of reduced adaptability. The choice between these settings represents a trade-off: $\lambda = 0.1$ offers more flexibility for learning complex preference patterns, while $\lambda = 1$ provides robustness and faster convergence. Both configurations successfully prevent the unbounded growth observed without regularization, making them viable choices depending on the specific requirements of the task.

The convergence patterns also reveal that inner thresholds (`zeta_2` and `zeta_3`, corresponding to $\zeta_{-1}$ and $\zeta_1$) stabilize more quickly than outer ones across all regularization settings, suggesting that distinguishing between moderate preference levels is learned before extreme preferences. The symmetric constraint is maintained throughout training, with negative thresholds mirroring their positive counterparts.

## G.2 TRAINING PHASES AND TRANSITIONS

When regularization is moderate (as with $\lambda = 0.1$), we observe a distinct multi-phase training behavior where thresholds evolve through discrete stages rather than converging smoothly. Figure 2 illustrates this phenomenon more clearly for a representative training run.

The training proceeds through several distinct phases, each characterized by relatively stable threshold values, with rapid transitions between them. This behavior is particularly pronounced for the outer thresholds (`zeta_0` and `zeta_5`, corresponding to $\zeta_{-3}$ and $\zeta_3$), which undergo several discrete jumps before reaching their final values. The inner thresholds exhibit similar but less dramatic phase transitions.

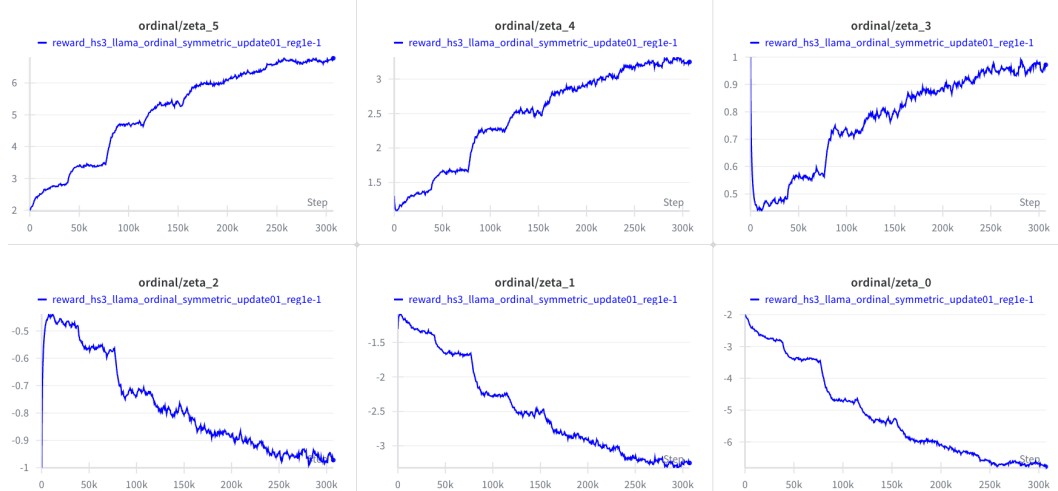

Figure 2: Multi-phase threshold evolution during training with moderate regularization ($\lambda = 0.1$). The thresholds exhibit step-like progression with distinct stable phases separated by rapid transition periods. Outer thresholds (`zeta_0` and `zeta_5`) show the most pronounced phase transitions.

This multi-phase behavior likely reflects the model's progressive refinement of its preference understanding. In early phases, the model learns coarse distinctions between preference levels, establishing rough boundaries. As training progresses and the model encounters more challenging examples that fall near existing boundaries, it triggers threshold adjustments to better capture the nuanced preference structure in the data. Each phase transition represents the model discovering a better configuration for separating preference levels, with the step-like nature suggesting that these improvements require coordinated changes across multiple thresholds rather than gradual local adjustments.

Importantly, during these transition phases—the periods of rapid threshold adjustment between stable phases—both the threshold values and the associated model weights exhibit increased instability. The reward predictions become less reliable as the model reconfigures its internal representation of preference levels. This instability makes checkpoints from transition phases unsuitable for evaluation, as they represent incomplete adaptations to new threshold configurations. In our experimental protocol, we therefore exclude checkpoints from these transition periods when selecting models for evaluation, focusing instead on checkpoints from stable phases where the thresholds have settled into consistent values. This filtering ensures that our reported results reflect the model's performance when operating with a coherent and stable preference structure rather than during periods of internal reorganization.

### G.3 REWARD VALUE STABILITY UNDER REGULARIZATION

Theorem 3.1 establishes that without regularization, rewards and thresholds can scale jointly to achieve arbitrarily low loss. Our regularization strategy addresses this by bounding threshold growth, which necessarily anchors the reward scale as well, since the two parameter sets are coupled through the loss function. Having demonstrated threshold convergence in Section G.1, we now empirically verify that reward values also remain stable under regularization.

Figures 3 and 4 show the evolution of reward values assigned to chosen and rejected responses during training and evaluation for the same NLL-Symmetric models analyzed in Section G.1. The empirical results confirm our theoretical predictions. Without regularization ($\lambda = 0$, red curves), reward values exhibit the unbounded growth predicted by Theorem 3.1. On the training set, chosen responses receive increasingly positive rewards while rejected ones receive increasingly negative rewards, with the separation growing continuously. On evaluation data, this divergence is even more pronounced, with chosen rewards growing increasingly positive and rejected rewards growing increasingly negative without bound, demonstrating the joint scaling of rewards and thresholds.

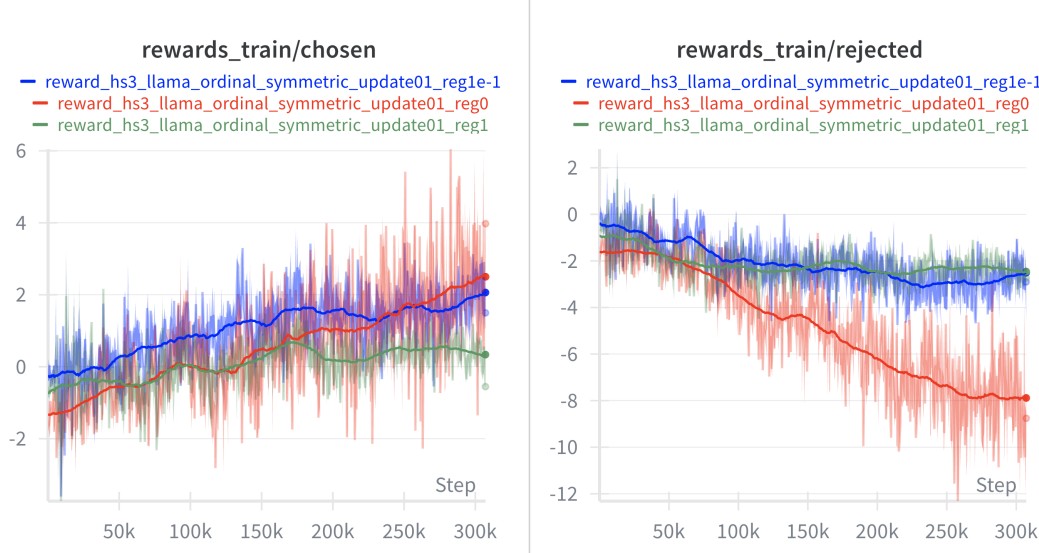

Figure 3: Training reward values for chosen and rejected responses across different regularization settings. Without regularization ($\lambda = 0$, red), rewards diverge significantly with chosen responses receiving increasingly large positive rewards and rejected ones receiving increasingly negative rewards. With regularization ($\lambda = 0.1$ in blue, $\lambda = 1$ in green), reward values remain bounded and stable throughout training.

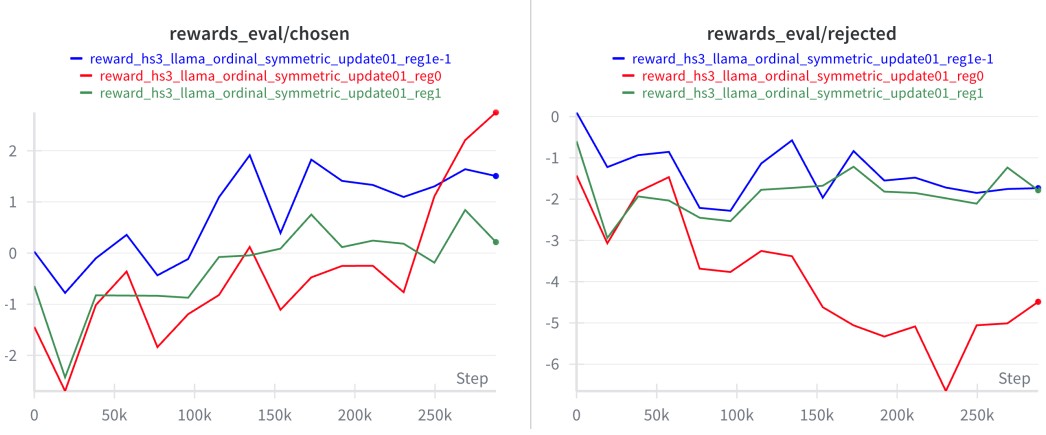

Figure 4: Evaluation reward values for chosen (left) and rejected (right) responses across different regularization settings. Without regularization (red), both chosen and rejected rewards diverge to extreme values. With regularization (blue and green), both reward types remain bounded and stable.

With threshold regularization ($\lambda = 0.1$ in blue and $\lambda = 1$ in green), reward values remain stable as theoretically expected. Training rewards converge to bounded values, with chosen and rejected responses receiving rewards around 0 to 2 and -2 to -4, respectively. On evaluation data, chosen rewards stabilize around 0 to 2 while rejected rewards stabilize around -1 to -2, remaining constant after initial convergence. The stronger regularization ($\lambda = 1$) produces slightly more conservative reward values compared to the moderate regularization ($\lambda = 0.1$), which is consistent with the faster threshold convergence observed in Figure 1.

These results validate that regularizing thresholds effectively anchors the reward scale through the coupling in the loss function. By preventing threshold growth, we simultaneously prevent rewards from scaling independently, ensuring convergence to a well-defined solution with stable and interpretable parameters for both rewards and thresholds.

# H  ORDINAL PERFORMANCE METRICS

While our main evaluation follows standard practice in reward modeling by focusing on binary preference accuracy—whether the preferred response receives a higher reward—this metric provides limited insight into the effectiveness of ordinal regression. Binary accuracy treats all correct predictions equally, failing to distinguish between a model that correctly predicts preference strength 1 versus strength 3, as long as the ranking is preserved. To assess whether our ordinal regression methods genuinely capture fine-grained preference structure, we introduce additional metrics that evaluate the accuracy of predicted preference strengths.

This appendix is organized into three parts. We first define the ordinal performance metrics used throughout our evaluation (§H.1). We then examine training dynamics, showing how these metrics evolve as our ordinal model learns (§H.2). Finally, we compare our joint training approach against post-hoc calibration of baseline methods, demonstrating that learning thresholds alongside reward parameters is essential for capturing ordinal structure (§H.3).

## H.1  METRIC DEFINITIONS

We evaluate ordinal performance using complementary metrics that measure different aspects of prediction quality:

**Mean Absolute Error (MAE).** The average absolute difference between predicted and actual preference strength levels

$$\text{MAE} = \frac{1}{n} \sum_{i=1}^{n} |z_{\text{pred}}^i - z_{\text{actual}}^i|,$$

where $z$ denotes the discrete ordinal level. Lower values indicate better performance, with perfect prediction yielding MAE = 0.

**Accuracy Within $k$.** We measure prediction accuracy allowing for different tolerance levels. Formally, Accuracy Within $k$ is defined as

$$\text{Acc}_k = \frac{1}{n} \sum_{i=1}^{n} \mathbb{1}[|z_{\text{pred}}^i - z_{\text{actual}}^i| \leq k],$$

where $\mathbb{1}[\cdot]$ is the indicator function. This metric captures the proportion of predictions that fall within $\pm k$ levels of the true preference strength.

For $k = 0$, this reduces to exact accuracy, the most stringent metric requiring perfect ordinal prediction beyond mere ranking preservation. For $k = 1$ and $k = 2$, we obtain near accuracy metrics that allow for minor deviations, recognizing that adjacent preference levels may have overlapping boundaries in human judgment. These metrics form a natural hierarchy $\text{Acc}_k \leq \text{Acc}_{k+1}$, providing increasingly relaxed measures of ordinal prediction quality. In our evaluation, we report MAE alongside $\text{Acc}_k$ for $k \in \{0, 1, 2\}$.

## H.2  TRAINING AND EVALUATION DYNAMICS

Figures 5 and 6 show the evolution of these metrics during training for NLL-Symmetric on Llama with HelpSteer3.

The training metrics show rapid improvement in the first 50k steps, with MAE dropping from above 2 to below 0.5 and exact accuracy rising from near 0% to above 60%. This corresponds to the period when thresholds are establishing their initial configuration (compare with Figure 2). The subsequent gradual improvement reflects fine-tuning of both thresholds and model weights.

On validation data, the model achieves approximately 55% exact accuracy and 85% accuracy within one level, with MAE stabilizing around 0.6. While exact accuracy is lower than on training data—expected given the challenge of generalizing fine-grained distinctions—the high near-accuracy demonstrates that the model successfully learns the ordinal structure rather than merely memorizing training examples. The fact that over 95% of predictions fall within two levels of the

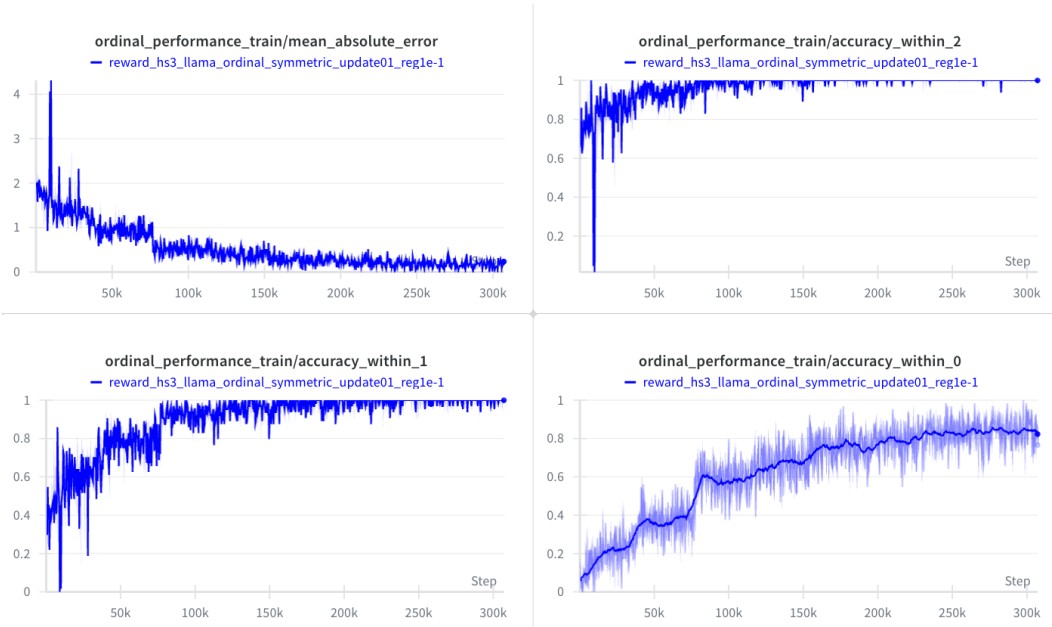

Figure 5: Ordinal performance metrics on the training set. MAE decreases rapidly in early training, stabilizing around 0.2. Exact accuracy reaches approximately 85%, while near-perfect accuracy (Within 1) approaches 95%, indicating strong ordinal prediction on training data.

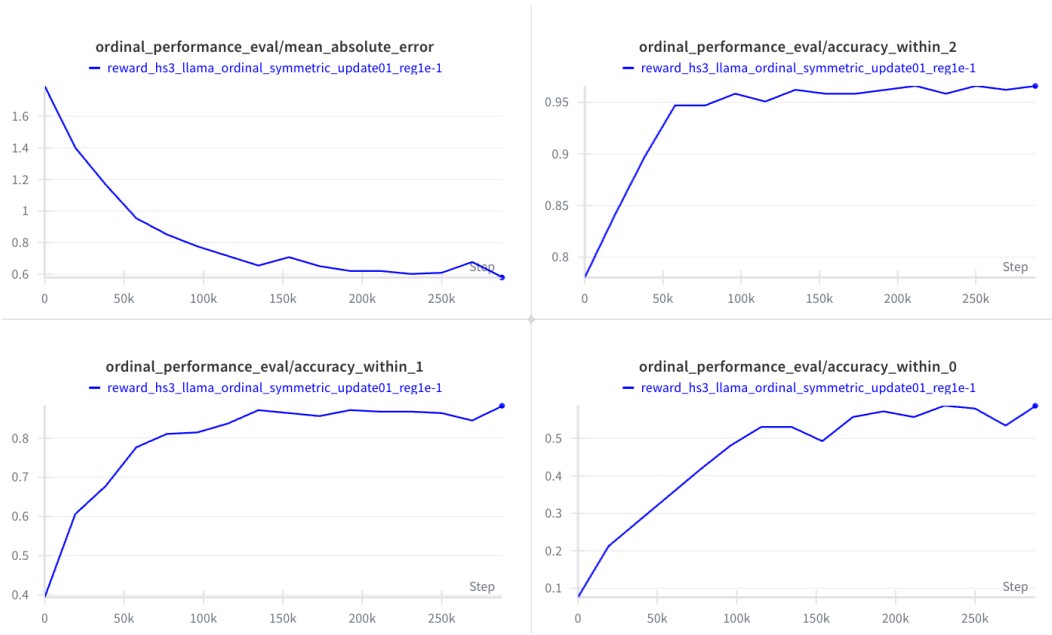

Figure 6: Ordinal performance metrics on the validation set. The model achieves approximately 55% exact accuracy and 85% accuracy within 1 level, demonstrating meaningful generalization of ordinal structure despite the more challenging validation distribution.

true value confirms that our ordinal regression approach captures meaningful preference strength information beyond what binary metrics reveal.

These results validate that our ordinal regression methods do more than preserve rankings; they learn to predict the degree of preference with reasonable accuracy, justifying the additional complexity of the ordinal framework over simpler binary approaches.

## H.3 COMPARISON WITH POST-HOC CALIBRATION

To validate that joint threshold learning is essential—rather than a convenience that could be replaced by post-hoc calibration—we compare our approach against baseline methods augmented with learned thresholds. Although baseline approaches (Margin BT, Scaled BT, Soft Label) do not natively produce ordinal predictions, we can enable ordinal evaluation through *post-hoc calibration*: after training each baseline model, we learn thresholds $\{\zeta_k\}$ that map reward differences to ordinal levels by optimizing the same ordinal negative log-likelihood objective used in our framework. This allows us to isolate the effect of threshold learning strategy: (1) models that learn thresholds jointly with reward parameters (our approach), versus (2) models that learn thresholds separately after reward model training (post-hoc calibration). If joint training offers no advantage, post-hoc calibration should achieve comparable ordinal performance.

### H.3.1 POST-HOC CALIBRATION PROCEDURE

After training a baseline reward model (Scaled BT, Margin BT, or Soft Label) for the standard 5 epochs on HelpSteer2, we freeze the model weights and learn thresholds $\{\zeta_{-3}, \zeta_{-2}, \zeta_{-1}, \zeta_1, \zeta_2, \zeta_3\}$ that map reward differences to ordinal predictions.

**Threshold Learning.** Given a trained reward model $r_\theta$ (with frozen parameters $\theta$), we compute reward differences $\Delta r_i = r_\theta(y_i^w) - r_\theta(y_i^l)$ for all training examples. Let $z_i \in \{-3, -2, -1, 0, 1, 2, 3\}$ denote the true ordinal strength for example $i$. We learn thresholds $\{\zeta_{-3}, \zeta_{-2}, \zeta_{-1}, \zeta_1, \zeta_2, \zeta_3\}$ (with $\zeta_{-4} = -\infty$ and $\zeta_4 = +\infty$, and no $\zeta_0$ since the interval $(\zeta_{-1}, \zeta_1)$ corresponds to $z = 0$) by minimizing the ordinal negative log-likelihood

$$\mathcal{L}_{\text{calib}}(\{\zeta_k\}) = -\sum_{i=1}^{n} \log p(y_i^w \succ_{z_i} y_i^l \mid x_i), \tag{50}$$

where the probability $p(y \succ_z y' \mid x)$ is given by the ordered logit model (following Equation 14 from the main text)

$$p(y \succ_z y' \mid x) = \begin{cases} \sigma(\zeta_z - \Delta r) - \sigma(\zeta_{z-1} - \Delta r) & \text{if } z \in \{-3, -2, -1\}, \\ \sigma(\zeta_{z+1} - \Delta r) - \sigma(\zeta_z - \Delta r) & \text{if } z \in \{1, 2, 3\}, \\ \sigma(\zeta_1 - \Delta r) - \sigma(\zeta_{-1} - \Delta r) & \text{if } z = 0. \end{cases}$$

To ensure monotonicity ($\zeta_{-3} < \zeta_{-2} < \zeta_{-1} < \zeta_1 < \zeta_2 < \zeta_3$), we parameterize thresholds using positive increments and optimize using gradient descent (100 epochs, learning rate 0.01).

**Ordinal Prediction.** After learning thresholds, we predict ordinal strength for a new example by finding which interval $(\zeta_k, \zeta_{k+1})$ the reward difference $\Delta r$ falls into (treating the interval $(\zeta_{-1}, \zeta_1)$ as corresponding to $z = 0$). Equivalently, we can compute

$$\hat{z} = \underset{z \in \{-3, \ldots, 3\}}{\arg\max} \ p(y \succ_z y' \mid x), \tag{51}$$

where $p(y \succ_z y' \mid x)$ is given by the ordered logit model above.

We explored two additional calibration methods—quantile-based (setting thresholds as empirical quantiles of reward differences) and classification-based (directly optimizing threshold positions to minimize MAE)—but found that ordinal NLL calibration consistently outperformed both. Therefore, we report only ordinal NLL calibration results for the baselines.

### H.3.2 EXPERIMENTAL SETUP

We evaluate all methods using the Llama-3.1-Tulu-3-8B model trained on HelpSteer2 for 5 epochs. For our ordinal regression method (NLL-Symmetric), thresholds are learned jointly during training. For baselines, we apply post-hoc calibration on the same training data, then evaluate on the held-out

test set. All models use identical architectures, training hyperparameters, and data splits, ensuring that differences in ordinal metrics reflect only the threshold learning strategy (joint versus post-hoc). We use seed 0 for reproducibility.

### H.3.3 RESULTS

Tables 4 and 5 present ordinal performance metrics for all methods on training and test sets, respectively.

Table 4: Ordinal performance metrics on HelpSteer2 training set (8,677 examples). NLL-Symmetric learns thresholds jointly during training, while baselines use post-hoc calibration with frozen reward models.

| Method | MAE ↓ | Acc@0 ↑ | Acc@1 ↑ | Acc@2 ↑ |
|---|---|---|---|---|
| Scaled BT (post-hoc) | 1.320 | 27.2% | 69.3% | 77.5% |
| Margin BT (post-hoc) | 0.994 | 29.2% | 77.8% | 95.7% |
| Soft Label (post-hoc) | 1.609 | 14.7% | 51.6% | 81.1% |
| NLL-Symmetric (joint) | **0.443** | **60.2%** | **95.7%** | **99.8%** |

Table 5: Ordinal performance metrics on HelpSteer2 test set (448 examples). Joint training (NLL-Symmetric) substantially outperforms post-hoc calibration across all metrics.

| Method | MAE ↓ | Acc@0 ↑ | Acc@1 ↑ | Acc@2 ↑ |
|---|---|---|---|---|
| Scaled BT (post-hoc) | 2.667 | 10.9% | 37.1% | 44.0% |
| Margin BT (post-hoc) | 2.181 | 16.1% | 48.7% | 59.6% |
| Soft Label (post-hoc) | 1.725 | 12.9% | 47.8% | 78.1% |
| NLL-Symmetric (joint) | **1.060** | **29.7%** | **72.5%** | **92.9%** |

**Joint Training Advantage.** On the training set, our jointly-trained ordinal model achieves substantially lower MAE (0.443) compared to the best post-hoc baseline (Margin BT: 0.994), representing a 55% reduction in error. More strikingly, exact accuracy (Acc@0) reaches 60.2% versus 29.2% for Margin BT, demonstrating that joint optimization enables the model to learn fine-grained preference distinctions that post-hoc calibration cannot recover.

The advantage persists and even amplifies on the test set. NLL-Symmetric achieves MAE of 1.060 compared to 1.725 for the best baseline (Soft Label), and exact accuracy of 29.7% versus 16.1% (Margin BT). The gap in Acc@1 is particularly notable: 72.5% for our method versus 48.7% for Margin BT, indicating that our model predicts preference strength within one level for nearly three-quarters of examples, while the best baseline achieves this for fewer than half.

**Baseline Comparison.** Among post-hoc calibrated baselines, Margin BT generally performs best on the training set (MAE: 0.994, Acc@1: 77.8%), likely because its margin-aware loss provides a better foundation for threshold learning. However, on the test set, Soft Label achieves the lowest MAE (1.725) and highest Acc@2 (78.1%), suggesting it generalizes differently than Margin BT. Scaled BT consistently underperforms, indicating that scaling rewards alone does not provide sufficient structure for accurate ordinal prediction.

### H.3.4 SUMMARY

These results validate that joint ordinal training provides substantial benefits over post-hoc calibration. The 38% reduction in test MAE (1.060 versus 1.725) and the doubled exact accuracy (29.7% versus 16.1%) demonstrate that learning thresholds alongside reward parameters is essential for capturing fine-grained preference structure. Post-hoc calibration, despite using the same probabilistic objective, cannot recover the ordinal information that emerges from joint optimization. This confirms that our ordinal regression framework does more than preserve rankings; it learns to predict

the degree of preference with meaningful accuracy that cannot be achieved through threshold fitting alone.

# I   ERROR MARGIN ANALYSIS

Beyond standard accuracy metrics, we analyze the *severity* of errors made by different models. When a reward model incorrectly ranks responses, assigning higher reward to the dispreferred response, the magnitude of this ranking violation provides insight into model calibration and confidence. Models that make large-margin errors may be particularly problematic in deployment, as they indicate strong but incorrect preferences.

## I.1   ERROR MARGIN DEFINITION

For each example in RewardBench consisting of a prompt $x$, chosen response $y_c$, and rejected response $y_r$, we define an error as occurring when $r_\phi(x, y_r) > r_\phi(x, y_c)$. The **error margin** for such misclassified examples is computed as

$$\text{Error Margin} = r_\phi(x, y_r) - r_\phi(x, y_c). \tag{52}$$

Larger values indicate that the model not only made an incorrect ranking decision but did so with high confidence, assigning substantially higher reward to the dispreferred response. This metric reveals whether a model's errors tend to be borderline cases (small margins) or egregious misrankings (large margins).

## I.2   EXPERIMENTAL COMPARISON

We compare the error margin distributions of two methods trained on HelpSteer2 using Llama-3.1-8B as the base model:

1. **Simple BT**: Standard Bradley-Terry loss treating all preferences as binary, completely ignoring ordinal strength information in the training data.

2. **NLL Ordinal Symmetric**: Our proposed negative log-likelihood ordinal regression approach with symmetric threshold constraints.

Both models are evaluated on the full RewardBench test set, and we analyze only the incorrectly ranked examples.

## I.3   RESULTS AND ANALYSIS

Figure 7 presents the error margin distributions for both methods. The contrast is striking across multiple dimensions:

**Error Frequency.**   Simple BT produces 433 errors on RewardBench, while NLL Ordinal Symmetric makes only 282 errors, showing a 35% reduction in total misclassifications. This improvement alone demonstrates the value of leveraging ordinal information.

**Error Severity.**   The mean error magnitude reveals an even more dramatic difference: Simple BT averages 3.827 in error margin, while NLL Ordinal Symmetric averages just 0.501. This is a reduction of over 87%. This indicates that our ordinal approach not only makes fewer mistakes but also makes *less severe* errors when it makes a mistake.

**Distribution Shape.**   The histograms reveal fundamentally different error characteristics:

- **Simple BT** (top panel) exhibits a heavy-tailed distribution with errors extending to margins as large as 20. The distribution shows substantial mass at moderate-to-large error margins (2-10), indicating frequent high-confidence misrankings. Even after the initial peak near zero, the distribution maintains considerable density across a wide range, suggesting systematic calibration issues.

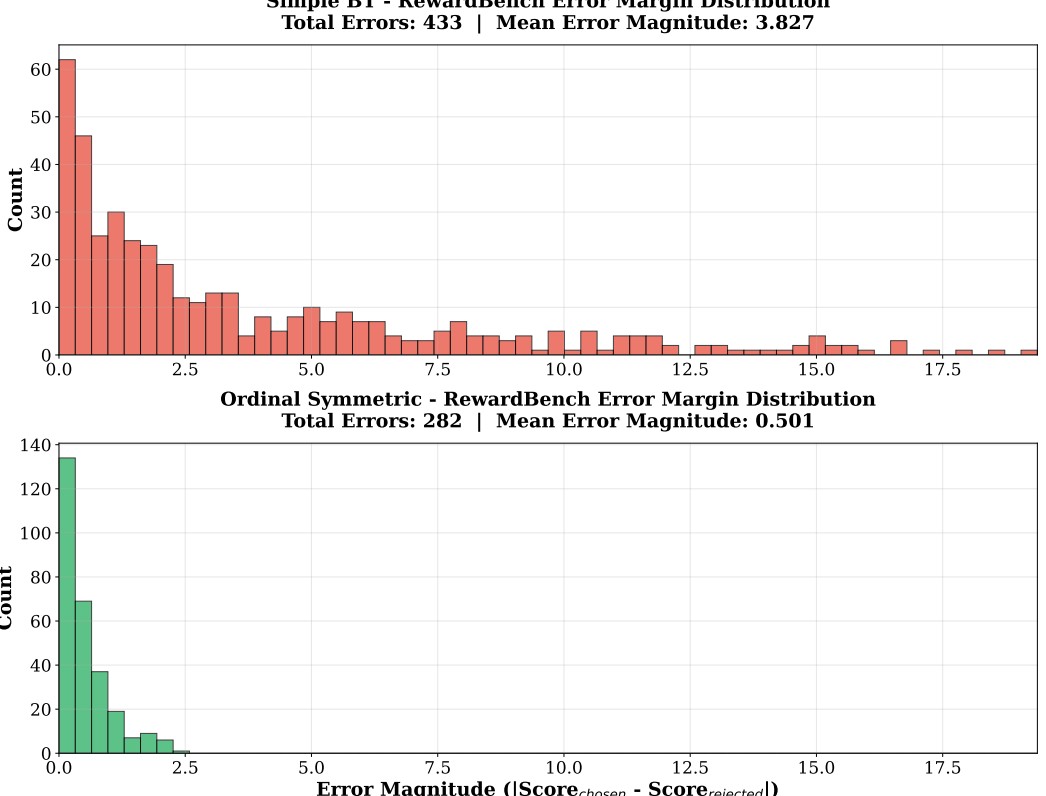

Figure 7: Error margin distributions comparing Simple BT (top) and NLL Ordinal Symmetric (bottom) on RewardBench. The ordinal method reduces both the frequency and severity of errors, with no errors exceeding margin 2.5 compared to Simple BT's errors extending to margin 20. The dramatic difference in mean error magnitude (3.827 vs 0.501) and distribution shape demonstrates that leveraging ordinal information improves not only ranking accuracy but also model calibration.

- **NLL Ordinal Symmetric** (bottom panel) displays a sharply concentrated distribution with nearly all errors confined to margins below 2.5. The distribution drops off rapidly after the peak near zero, with virtually no errors exceeding margin 3. This tight concentration indicates that when the ordinal model errs, it does so with low confidence. The reward differences are small, suggesting genuine ambiguity rather than systematic misunderstanding.

**Maximum Error Magnitude.** Perhaps most significantly, NLL Ordinal Symmetric produces no errors with margins exceeding approximately 2.5, while Simple BT generates errors with margins up to 20. These large-margin errors in Simple BT represent cases where the model is highly confident in an incorrect preference, potentially the most dangerous failure mode in practice.

## I.4 Implications

These results demonstrate that ordinal regression provides benefits beyond simple accuracy improvements. By learning from fine-grained preference strength information, the model develops better calibration and becomes more hesitant to make strong predictions when preferences are subtle or ambiguous. The dramatic reduction in large-margin errors suggests that ordinal methods produce more reliable reward signals for downstream applications like reinforcement learning, where high-confidence incorrect rewards could severely misguide policy optimization.

## J ROBUSTNESS TO LABEL NOISE

Real-world preference datasets inevitably contain annotation errors due to annotator disagreement, subjective judgment, and cognitive biases. Recent empirical work has documented the prevalence and complexity of such disagreement: for instance, Alipour et al. (2026) show that even experienced human moderators exhibit substantial disagreement when labeling the same content, highlighting the inevitability of noisy labels in human annotation pipelines. Understanding how reward models behave under label noise is crucial for practical deployment. While binary preference models face noise as simple flips (preferred $\leftrightarrow$ dispreferred), ordinal regression confronts a richer noise structure where labels can be corrupted in varying degrees. Recent work has highlighted the importance of noise robustness in preference learning: Afzali et al. (2025) address content-dependent multi-source noise in preference optimization through multi-objective optimization, while Im and Li (2025) provide theoretical generalization guarantees under noisy feedback, showing that moderate noise rates need not prevent effective learning if the data is well-structured. Here we investigate whether our ordinal regression framework exhibits robustness to different patterns of label corruption.

### J.1 NOISE MODELS

We evaluate robustness under the following two noise paradigms that represent different real-world annotation failure modes:

**Systematic Shift Noise.** This models annotators who consistently apply incorrect thresholds when assigning preference strength. For instance, an annotator might systematically overestimate or underestimate preference intensity. Given a true ordinal label $z \in \{-K, \ldots, K\}$, we corrupt it by shifting one level up or down with equal probability, i.e.,

$$z_{\text{noisy}} = \begin{cases} \min(z + 1, K) & \text{with probability } 0.5, \\ \max(z - 1, -K) & \text{with probability } 0.5. \end{cases} \tag{53}$$

This shifts labels by one ordinal level while respecting the boundaries of the ordinal scale. The noise preserves the general magnitude of preference (a strong preference remains relatively strong), making it representative of calibration errors rather than complete confusion.

**Random Label Noise.** This models annotators who occasionally assign a completely arbitrary label, perhaps due to fatigue, misunderstanding, or lack of attention. We replace the true label with a uniformly random draw from $\{-K, \ldots, K\}$, i.e.,

$$z_{\text{noisy}} \sim \text{Uniform}(\{-K, -K + 1, \ldots, K - 1, K\}). \tag{54}$$

This represents catastrophic annotation failures where the assigned label bears no relationship to the true preference, making it a much more challenging noise setting.

For each noise type, we evaluate at corruption rates of 25%, 50%, 75%, and 100%, representing increasingly severe label corruption. At each corruption rate, the specified percentage of training examples has their labels replaced according to the noise model. The extreme case of 100% corruption provides insight into the theoretical limits of model robustness.

### J.2 EXPERIMENTAL CONFIGURATION

We train NLL-Symmetric models (our best-performing method from Section 4) on Llama-3.1-8B with HelpSteer2 under each noise condition. All hyperparameters remain identical to the noise-free setting, including learning rates and regularization strength. Models are trained for 5 epochs and evaluated on the clean (noise-free) validation set to measure their ability to recover true preferences despite training on corrupted labels.

### J.3 BENCHMARK PERFORMANCE UNDER NOISE

Tables 6 and 7 present comprehensive evaluation results on RewardBench and RM-Bench, respectively.

Table 6: RewardBench performance under different noise conditions. All models use NLL-Symmetric with Llama-3.1-8B trained on HelpSteer2.

| Noise Setting | Chat | Chat Hard | Safety | Reasoning | Average |
|---|---|---|---|---|---|
| No Noise | 0.941 | 0.728 | 0.897 | 0.804 | 0.843 |
| Shift 25% | 0.953 | 0.697 | 0.892 | 0.771 | 0.828 |
| Shift 50% | 0.930 | 0.728 | 0.895 | 0.764 | 0.829 |
| Shift 75% | 0.930 | 0.726 | 0.850 | 0.726 | 0.808 |
| Shift 100% | 0.955 | 0.693 | 0.853 | 0.881 | 0.846 |
| Random 25% | 0.930 | 0.678 | 0.885 | 0.722 | 0.804 |
| Random 50% | 0.894 | 0.636 | 0.878 | 0.619 | 0.757 |
| Random 75% | 0.872 | 0.548 | 0.743 | 0.724 | 0.722 |
| Random 100% | 0.542 | 0.441 | 0.480 | 0.668 | 0.532 |

Table 7: RM-Bench performance under different noise conditions. All models use NLL-Symmetric with Llama-3.1-8B trained on HelpSteer2.

| Noise Setting | Chat | Math | Code | Safety | Hard | Normal | Easy | Average |
|---|---|---|---|---|---|---|---|---|
| No Noise | 0.669 | 0.558 | 0.515 | 0.873 | 0.418 | 0.681 | 0.862 | 0.654 |
| Shift 25% | 0.718 | 0.575 | 0.518 | 0.882 | 0.446 | 0.702 | 0.872 | 0.673 |
| Shift 50% | 0.683 | 0.557 | 0.517 | 0.919 | 0.427 | 0.698 | 0.883 | 0.669 |
| Shift 75% | 0.661 | 0.569 | 0.514 | 0.878 | 0.418 | 0.684 | 0.865 | 0.655 |
| Shift 100% | 0.628 | 0.556 | 0.527 | 0.878 | 0.371 | 0.684 | 0.886 | 0.647 |
| Random 25% | 0.706 | 0.556 | 0.518 | 0.839 | 0.430 | 0.677 | 0.857 | 0.655 |
| Random 50% | 0.633 | 0.557 | 0.500 | 0.826 | 0.346 | 0.655 | 0.886 | 0.629 |
| Random 75% | 0.554 | 0.531 | 0.489 | 0.815 | 0.405 | 0.619 | 0.768 | 0.597 |
| Random 100% | 0.426 | 0.485 | 0.498 | 0.398 | 0.371 | 0.442 | 0.542 | 0.452 |

The benchmark results reveal striking differences in robustness between the two noise types. Performance remains remarkably stable under systematic shift noise across both benchmarks. On RewardBench, the average accuracy ranges from 0.808 to 0.846 across all shift noise levels, staying within the noise level or slightly below the average accuracy of the no-noise baseline of 0.843. Similarly, on RM-Bench, performance varies only from 0.647 to 0.673. These small variations are not meaningfully different from the baseline, demonstrating strong robustness to systematic calibration errors.

Random label noise presents a more challenging scenario with graceful degradation. At 25% corruption, performance remains competitive with the no-noise baseline (0.804 on RewardBench, 0.655 on RM-Bench), indicating strong robustness to moderate random noise. At 50% corruption, where half the training labels are completely arbitrary, the model achieves 0.757 and 0.629 respectively, representing moderate degradation from baseline. The degradation accelerates at 75% noise and becomes severe at 100% corruption. Notably, since 50% accuracy represents chance-level performance on these benchmarks, the 100% random noise results (0.532 and 0.452) confirm that the model learns essentially nothing when trained entirely on random labels, as expected.

Certain evaluation categories exhibit differential sensitivity to noise. The Safety category maintains high scores even under heavy noise (0.743-0.895 on RewardBench across all conditions except 100% random), suggesting that safety-related preferences have clearer signal that survives label corruption. Conversely, Chat Hard and Reasoning categories show more pronounced degradation under random noise, indicating these tasks may benefit more from clean, high-quality annotations.

## J.4 TRAINING AND EVALUATION DYNAMICS

Figures 8 through 11 provide detailed insight into how label noise affects learning dynamics and generalization.

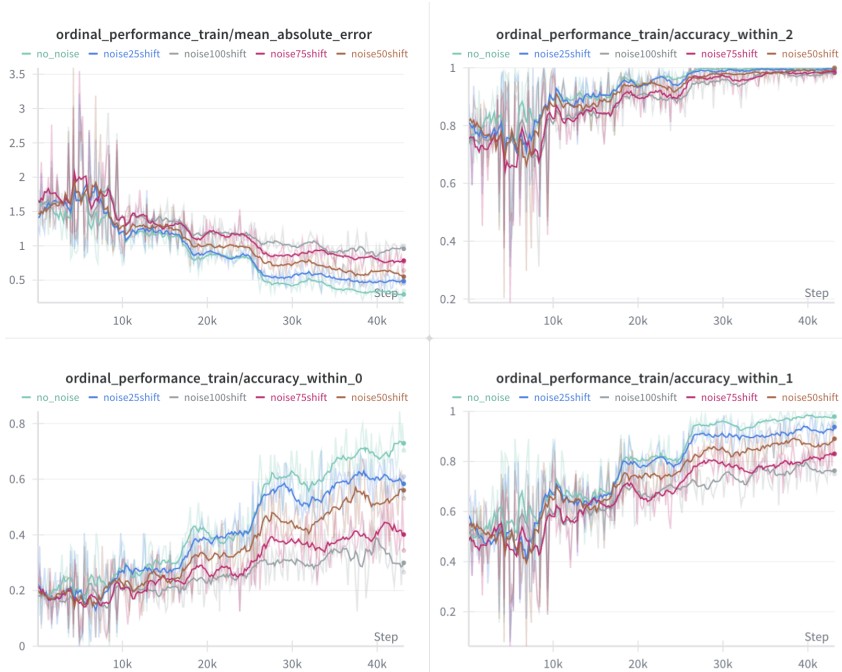

Figure 8: Training dynamics under systematic shift noise (left: mean absolute error, right: accuracy metrics). Training performance degrades with increasing noise levels, indicating the model does not overfit to corrupted labels.

**Systematic Shift Noise.**    Comparing Figures 8 and 9 reveals a crucial pattern: training performance degrades with increasing shift noise levels, while evaluation performance remains nearly constant across all noise conditions. This contrast demonstrates that the model is not overfitting to the noise. If the model were memorizing the corrupted training labels, we would observe high training accuracy accompanied by poor validation performance. Instead, the degraded training curves indicate the model resists fitting to the shifted labels, while the stable evaluation curves show it successfully recovers the true preference structure. This behavior confirms genuine robustness rather than mere memorization, as the learned thresholds effectively compensate for systematic biases in the training labels.

**Random Label Noise.**    Figures 10 and 11 show systematic degradation as noise level increases, with both training and evaluation performance declining progressively. At 100% corruption, the model fails to learn anything meaningful, as evidenced by training metrics that barely improve and evaluation performance at chance level. This type of noise is fundamentally more severe than systematic shifts because it destroys the ordinal structure entirely rather than merely miscalibrating it. The parallel degradation in training and evaluation metrics confirms that the model cannot extract useful signal from purely random labels.

The training and evaluation dynamics support the findings in the benchmark tables, demonstrating that ordinal regression exhibits strong robustness under reasonable noise conditions. The framework handles systematic calibration errors with minimal performance impact, while degrading gracefully under moderate random corruption. These properties make ordinal regression well-suited for practical deployment where annotation quality may vary but is unlikely to be entirely random.

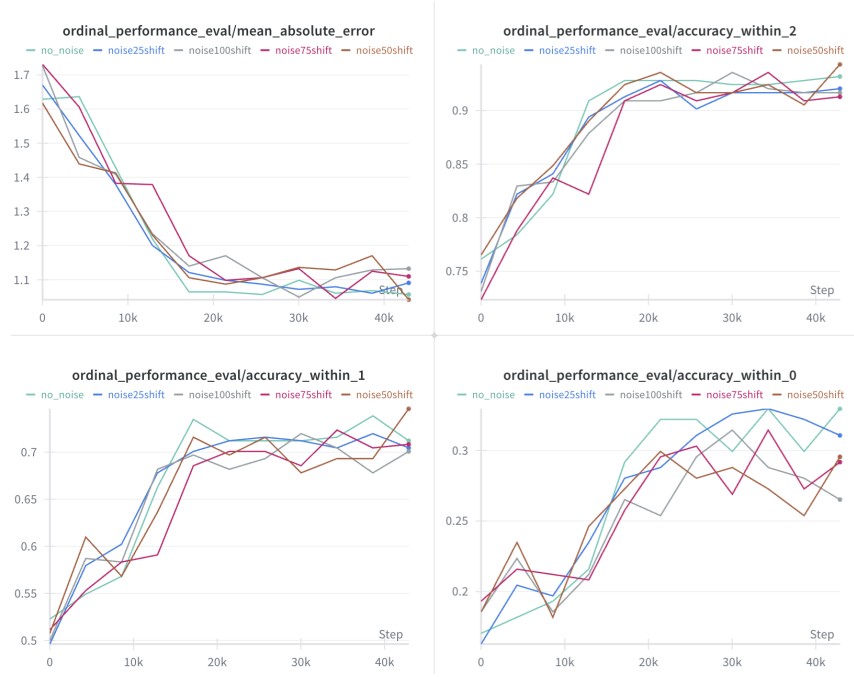

Figure 9: Evaluation performance under systematic shift noise on clean validation data (left: mean absolute error, right: accuracy metrics). All noise levels achieve similar validation performance, demonstrating robustness to systematic label shifts.

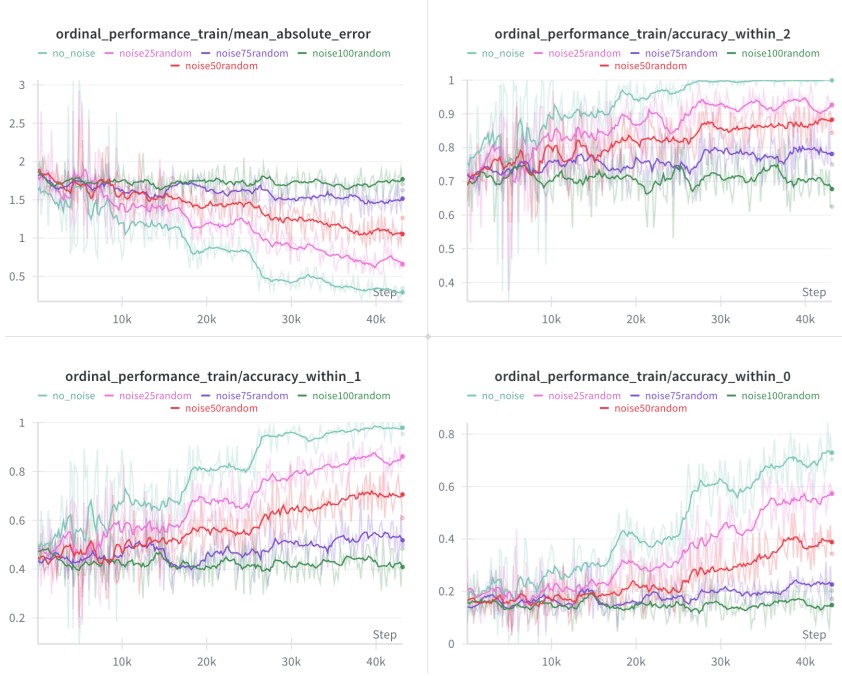

Figure 10: Training dynamics under random label noise (left: mean absolute error, right: accuracy metrics). Training performance degrades systematically with noise level, with 100% corruption preventing meaningful learning.

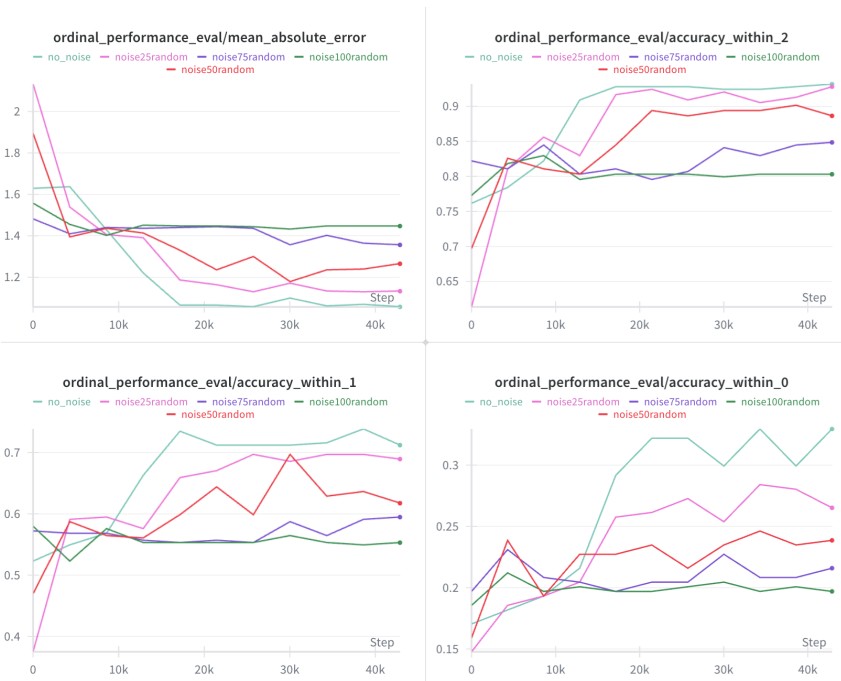

Figure 11: Evaluation performance under random label noise on clean validation data (left: mean absolute error, right: accuracy metrics). Performance degrades progressively with noise level, with 100% corruption resulting in near-random performance.

