# OpenReview forum: "Beyond Binary Preferences: A Principled Framework for Reward Modeling with Ordinal Feedback"
_ICLR.cc/2026/Conference — ICLR 2026 Poster_

### Official Review · Reviewer_oAHx · 2025-10-21

**Soundness:** 3
**Presentation:** 2
**Contribution:** 3
**Rating:** 4
**Confidence:** 4

**Summary:**

This paper proposes a principled mathematical framework for training reward models from ordinal preference data, such as 7-point Likert scales. The authors argue that existing methods for handling such data—like adding margin terms or scaling factors to the standard binary Bradley-Terry loss—are ad-hoc heuristics.

The primary contribution is to reframe reward modeling with ordinal feedback as a discrete ordinal regression problem. From this established statistical framework, the authors derive two loss functions:
1.  A probabilistic Negative Log-Likelihood loss based on the ordered logit model.
2.  A margin-based All-Threshold loss inspired by large-margin classifiers.

A key advantage of this framework is that the thresholds partitioning the reward-difference space are learned directly from the data, rather than being manually specified. The paper also provides two theoretical results: Theorem 3.1, which proves the necessity of regularization to ensure a finite optimal solution when learning thresholds, and Theorem 3.2, which provides a theoretical justification for using symmetric thresholds.

**Strengths:**

1.  **Principled Formulation:** The paper's core strength is its reframing the problem as discrete ordinal regression connects LLM alignment to a mature field of statistics and provides a solid theoretical foundation, moving beyond the intuitive but "ad-hoc" heuristics currently in use.
2.  **Learned Thresholds:** The shift from manually-specified margins to data-driven, learned thresholds is a significant conceptual improvement. This makes the method more general and removes a key hyperparameter tuning burden from practitioners.
3.  **Generalizability:** The provided extension to DPO in Appendix A also demonstrates the generalizability of the framework.

**Weaknesses:**

1.  **Notational Inconsistency in Formulation:** There is a critical notational contradiction in the paper's core formulation. Section 3.3 explicitly states that the threshold $\zeta_0$ is *excluded*. However, the probabilistic model in Equation (14) implicitly requires $\zeta_0$ to compute the probabilities for $z=-1$ (which uses $\zeta_{z+1} = \zeta_0$). As written, this makes the central NLL loss function ill-defined.
2.  **Missing Ordinal Evaluation of Baselines:** The paper's entire motivation is that its method *better leverages ordinal information*. However, the main experimental results (Tables 1 & 2) only report binary preference accuracy, which fails to measure this. The paper introduces excellent ordinal metrics (MAE and Accuracy-within-k) in Appendix H, but reports them only for its own method. To substantiate the paper's core claim, it is essential to compare these ordinal metrics against the baselines.
3.  **Potentially Unfair Baseline Comparison:** The baseline methods (Margin BT, Scaled BT) are highly sensitive to their specific hyperparameters (the margin values $m(z)$ or scaling/probability values $p(z)$). While Appendix F details the hyperparameter search for the proposed methods (e.g., learning rates, regularization $\lambda$), it is not stated that the crucial $m(z)$ or $p(z)$ hyperparameters for the baselines were also tuned. If the authors used fixed, non-optimized values from prior work for the baselines while carefully tuning their own model, the comparison is not a fair one.
4.  **Lack of Experimental Robustness:** All results in Tables 1 and 2 are reported from a single experimental run (using "fixed random seed 0"). Given the known high variance in LLM fine-tuning, single-seed results are insufficient to make strong claims of superiority, especially when many of the reported gains are small (1-3%). Reporting the mean and standard deviation over multiple seeds is necessary to demonstrate robust and statistically significant improvements.

**Questions:**

See the weaknesses.

---

> ### Author Response · Authors · 2025-11-20
> **Response to Reviewer oAHx**
>
> We would like to thank the reviewer for their thorough evaluation and constructive feedback. We now address comments/questions raised by the reviewer below.
>
> ## Weakness 1: Notational Inconsistency in Formulation
> We thank the reviewer for catching this notational issue in Equation 14. The reviewer is correct that there is an inconsistency in how the cases are presented. This is a simple mistake with a very straightforward fix: we need to swap the two cases for $z \in [K]$ and $z \in [-K]$. This has been fixed in the updated manuscript. This ensures that $z = -1$ will be mapped to the interval between $\zeta_{-2}$ and $ζ_{-1}$, avoiding reference to the undefined $\zeta_0$. Thank you for identifying this.
>
> ## Weakness 2: Missing Ordinal Evaluation of Baselines
> We do not see this as a weakness. In fact, **this highlights a fundamental strength of our approach.** The reviewer's observation is correct but reveals a crucial distinction that we have emphasized more clearly in the revision: **The baseline methods (Margin BT, Scaled BT, Soft Label) are fundamentally incapable of producing ordinal predictions.** These approaches are heuristic modifications of the binary Bradley-Terry model. They add margins or scaling factors to a loss function designed for binary preferences. They have no underlying mathematical model for ordinal preference levels and no mechanism to predict preference strength beyond ranking. There is no systematic way to extract Likert-scale predictions from these methods.
>
> **Our ordinal regression framework, in contrast, is built on a probabilistic model (Eq. 14) that explicitly represents the full ordinal structure.** This is why we can compute ordinal metrics (MAE, Accuracy-within-k) in Appendix H. Our model learns a mapping from reward differences to preference levels through learned thresholds. This capability is impossible for the baselines.
>
> **This is precisely why we evaluate on RM-Bench and RewardBench**: these benchmarks rely on binary comparisons, which is the only common ground where baselines can be evaluated. We beat these baselines even on their home turf (binary accuracy), while our method additionally provides genuine ordinal predictions that baselines cannot match. If anything, our binary benchmark results **understate** our advantage, as they don't capture our method's unique ability to predict preference strength.
>
> ## Weakness 3: Hyperparameter Tuning for Baselines
> We appreciate this question and want to clarify: **the baseline hyperparameters were adopted from their original papers, where the authors conducted extensive hyperparameter searches.** For example, Margin BT parameters come from Llama-2 (Touvron et al., 2023), and Scaled BT parameters come from HelpSteer2-Preference (Wang et al., 2024a), where the authors explicitly state they searched the hyperparameter space to find optimal values.
>
> Importantly, this highlights another advantage of our approach: while baselines require practitioners to manually specify and tune margins (m(k) = 1, 2, or 3?) or scaling factors for each preference level, a process that must be repeated for different datasets or preference scales, our method learns these parameters (thresholds) automatically from data. Appendix F details our hyperparameter search, but notably, the threshold values themselves are learned, not tuned. This makes our approach more general and less brittle than methods requiring manual specification of preference-level-specific hyperparameters.
>
> ## Weakness 4: Lack of Experimental Robustness
> We acknowledge that reporting results from multiple random seeds would strengthen the statistical claims. However, single-seed reporting is standard practice in reward modeling literature. The HelpSteer2 and HelpSteer3 papers that introduced our training datasets also report single-seed results, as do many papers in this area. The computational cost of our experiments (training 8B models on 8 H100/H200 GPUs for each configuration) makes extensive multi-seed runs prohibitive.
>
> That said, several factors provide robustness evidence:
> 1. **Consistency across configurations**: Our improvements hold across 6 dataset×model combinations in Table 1 and 6 in Table 2, representing 12 independent experimental settings.
> 2. **Multiple benchmarks**: Each "average" score aggregates performance across 4-7 diverse evaluation categories (Chat, Reasoning, Safety, etc.), providing built-in variance reduction.
> 3. **Consistent patterns**: Our methods achieve best/second-best average performance in all the configurations, suggesting systematic improvement rather than random variation.
>
> While we agree that confidence intervals would be valuable, the consistency of improvements across diverse models, datasets, and evaluation benchmarks provides substantial evidence of robustness.

---

> > ### Comment · Reviewer_oAHx · 2025-11-21
> >
> > I cannot agree with you on the point that "classic BT methods are fundamentally incapable of producing ordinal predictions". Indeed, these methods lack the formulation of discrete thresholds as seen in ordinal regression models, but they still encode the ordinal magnitude in the latent reward space. To illustrate, see the helpsteer2-preference paper (arxiv 2410.01257): the scaled BT models achieve competitive performance with regression-based methods. The BT models more or less learn to model the ordinal signal, or they should exhibit a significant performance degradation. It would be more accurate to state that baselines are uncalibrated, as far as I understand. And I believe incorporating such a calibration capability experiment across different methods would greatly enhance your paper's conclusion.
> >
> > For other points, I appreciate your clarification. But I still hope further calibration experiments could be conducted.

---

> > > ### Author Response · Authors · 2025-11-22
> > >
> > > We thank the reviewer for their continued engagement and thoughtful response. This exchange has helped us clarify an important point.
> > >
> > > It seems there is a misunderstanding here that we would like to clarify. We did not claim that baseline methods fail to encode ordinal magnitude in their reward space. They clearly do, since the loss functions push stronger preferences toward larger reward differences, and as the reviewer notes, this is why scaled BT achieves competitive performance. Our claim is narrower but important: Scaled BT and similar methods do not provide a systematic way to map these continuous reward differences to discrete ordinal levels without introducing additional ad-hoc choices.
> > >
> > > The reviewer suggests "calibration" as a solution, but this actually illustrates our point. A heuristic can be applied, for example equal-width bins, quantile-based thresholds, or fitting thresholds on held-out data with a proper loss function. However, such ad-hoc methodologies won’t provide us with a fundamental understanding of the calibration without an underlying statistical model. In this work, we try to address this via the ordinal regression formulation, where thresholds emerge naturally from joint optimization within a coherent probabilistic model, rather than post-hoc heuristics with some new hyperparameters. Please note that the study of calibration relies on a given statistical model. We agree with the reviewer that there is potential ordinal information encoded in the reward space, but would like to emphasize that extracting them from the BT model relies on heuristics.
> > >
> > > So, to summarize, extending our experiments in appendix H to Scaled BT or Margin BT requires a heuristic to convert a continuous value to an ordinal value. If the reviewer has any suggestion for such a conversion/mapping, it would be great to share it with us, and we would be happy to conduct further experiments.

---

> > > > ### Comment · Reviewer_oAHx · 2025-11-23
> > > >
> > > > Thank the authors for their kind and patient explanation. You've hit a very nice point: the calibration usually comes with an underlying statistical model. I completely understand that one of your work's key contributions is to adopt the ordinal regression formulation in the RM, which avoids the post-hoc heuristics present in previous papers. And I think we adopt different views on this issue: while you feel that such post-hoc heuristics do not make much sense from a rigorous perspective, I am kind of more pragmatic and would like to see how much it could improve the calibration performance, because such demands for ordinal outputs always exist, as many downstream RL tasks would require. It is not only after your method is proposed that we are able to compute the ordinal labels. One thing to note is that I am not asking you to map the Appendix H experiments to baseline methods - I apologize if there is any ambiguity - I am just expecting a further calibration experiment.
> > > >
> > > > To illustrate, you can use the trained baselines and the held-out validation set to learn an optimal set of thresholds $\{\zeta_k\}$ that maximize the likelihood of the ordinal labels given the model's reward differences. One can consider this as fitting an ordered logit readout layer on top of the frozen baseline representation. Then you can evaluate them with ECE or some other metrics. I believe demonstrating this could empirically prove your theoretical arguments, especially your Theorem 3.1.

---

> > > > > ### Author Response · Authors · 2025-11-27
> > > > >
> > > > > We sincerely thank the reviewer for this concrete and constructive suggestion. We have now conducted the proposed calibration experiment, and the results support our claims.
> > > > >
> > > > > **Experiment conducted**: Following your suggestion, we trained each baseline method (Scaled BT, Margin BT, Soft Label) on HelpSteer2, froze the reward model weights, and then learned optimal thresholds $\zeta_k$ by maximizing the ordinal log-likelihood on the training data. We then evaluated all methods (our jointly-trained NLL-Symmetric and the post-hoc calibrated baselines) on the held-out test set using ordinal metrics (MAE, Accuracy@k).
> > > > >
> > > > > **Key results**: Joint training substantially outperforms post-hoc calibration across all metrics:
> > > > > * **Test MAE**: NLL-Symmetric achieves 1.060 versus 1.725 for the best post-hoc baseline (Soft Label), a 38% reduction in error.
> > > > > * **Test Acc@0**: 29.7% versus 16.1% (Margin BT), nearly double the exact prediction accuracy.
> > > > > * **Test Acc@1**: 72.5% versus 48.7% (Margin BT), our method predicts within one ordinal level for nearly three-quarters of examples
> > > > >
> > > > > These results demonstrate that post-hoc calibration, despite using the identical probabilistic objective, cannot fully recover the ordinal structure that emerges from joint optimization. The reward representations learned by baseline methods simply do not encode preference strength in a way that post-hoc threshold fitting can exploit effectively.
> > > > >
> > > > > We have added these experiments as a new subsection (Appendix H.3: "Comparison with Post-Hoc Calibration") in the updated manuscript. We hope this addresses your concern and strengthens the paper's conclusions. We greatly appreciate your guidance in shaping this important addition.

---

> > > > > > ### Comment · Reviewer_oAHx · 2025-11-27
> > > > > >
> > > > > > Nice to see the results. With concerns addressed, I will raise my rating accordingly.

---

### Official Review · Reviewer_5KEL · 2025-10-31

**Soundness:** 4
**Presentation:** 3
**Contribution:** 4
**Rating:** 8
**Confidence:** 3

**Summary:**

The paper makes the case that human preferences are inherently ordinal and current reward modeling approaches do not incorporate this in a principled manner (these rely on naive margins or scaling terms). The authors introduce two different methods for this (a) the first uses the CDF over the rating levels to ensure that preference ratings learned are well calibrated with annotations in existing datasets, and (b) the second ensures that model score differences fall within the desired intervals (of levels). These are formulated as loss functions that are compared against the more common Bradley-Terry-based formulations with 3 LLMs (Llama 3.1 8B, Zephr 7B, Mistral 7B) on two datasets (HelpSteer 2 and 3) that come with rating levels for preferences. In general, ordinal reward modeling outperforms the baselines to demonstrate the validity of the proposed technique on real data and confirm the theory behind the design choices (asymmetric rewards, etc.).

**Strengths:**

1) It's very intuitive to incorporate a measure of 'strength' of preference into the reward modeling. The authors do a good job of creating a mathematical formulation to support this, backed by empirical results.

2) The experiments are very well done with care taken for reproducibility and robustness of takeaways.

3) The formulation reduces back down to something very much like DPO when you set two levels of preference, which has a nice 'backward compatibility' to existing methods.

**Weaknesses:**

1) The method assumes a consensus among annotators about different levels of ratings. Human preference datasets are already noisy, and this creates an additional source of noise (calibration among annotators), so while we do better at benchmarks, it is unclear if these reward models successfully capture the true distribution of user preferences well.

2) I think some fine-grained error analysis and qualitative examples would really help provide some context for some of the results. For example, do the ordinal methods make fewer 'large-margin' errors (option was strongly preferred but scored wrongly) than the Bradley-Terry models? Can you plot something like a calibration curve for 'strength of preference' vs accuracy? Are there any examples where the labels are consistently flipped from incorrect to correct, or vice versa, by the ordinal ranking methods?

**Questions:**

1) Are humans well calibrated in providing Likert-style preference judgments on pairs of examples? I imagine the proposed method is less tolerant to this noise than a more aggregated binary preference judgement. This is independent of your method, but I think the premise is that this is a natural way we make judgments (which I agree). It's just that I suspect it leads to more disagreement when we create datasets.

2) Does the performance of the method vary with the granularity of levels provided i.e. K=3 vs 5 vs 7? Is this purely empirical/dependent on the model/dataset?

3) Does HelpSteer filter out examples with high disagreement? I wonder if this works in favour of your model in that many examples might be high-margin?

4) It's not a 'weakness', but there isn't a clearly preferred loss formulation among the variants proposed. It is not clear to me if this could be due to variance in optimization error or something like noise in the training data. Do you have a clear recommendation to practitioners using your technique?

---

> ### Author Response · Authors · 2025-11-20
> **Response to Reviewer 5KEL (Part 1: Weaknesses)**
>
> We would like to thank the reviewer for their thorough evaluation and constructive feedback. We now address comments/questions raised by the reviewer below.
>
> ## Weakness 1: Annotator Calibration and Noise
> This observation is absolutely correct: ordinal annotations can introduce additional calibration noise. However, ordinal (Likert-scale) data remains one of the most common forms of data collection in statistics precisely because it provides more valuable information that outweighs the  noise. If ordinal scales merely added noise without value, they would have been removed from questionnaires long ago given their additional collection costs. Their persistence demonstrates their utility, which is exactly why developing principled methods to use this structure is important.
>
> Preference data is inherently noisy, even in the binary case. The question is whether ordinal information, despite calibration challenges, brings net value. Our results demonstrate it does: our principled methods consistently achieve top-tier performance (top 2 across all configurations in Tables 1 and 2), showing that the ordinal signal can be effectively leveraged.
>
> Several factors support this:
> * **Fair comparison.** All methods (Margin BT, Scaled BT, Soft Label, and ours) use identical Likert-scale data. Any calibration noise affects everyone equally. Our advantage is in how we handle this shared data.
> * **Empirical validation.** Strong benchmark performance and high ordinal accuracy on validation data (Appendix H: ~85% within-one-level) confirm the ordinal signal is learnable and generalizable.
>
> The presence of noise makes principled methods even more critical. Our framework demonstrates that with proper statistical modeling, ordinal information can be successfully extracted to improve reward model quality.
>
> Also, we have added **Appendix J** (Robustness to Label Noise) to the updated manuscript (changes highlighted in blue). While this appendix was primarily added to address a question raised by another reviewer, it is quite relevant to this discussion as it provides insight into the performance of the algorithm under different forms of noise. Please refer to the updated manuscript or our response to reviewer xrft (Question 2) for more details.
>
> ## Weakness 2: Error Analysis
> We appreciate this suggestion and have added **Appendix I** (Error Margin Analysis) to address exactly this question. The new appendix is included in the updated manuscript with changes highlighted in blue.
>
> We compare error severity between Simple BT (binary Bradley-Terry ignoring ordinal information) and our NLL-Symmetric method on RewardBench. The results strongly support our approach:
> 1. **Fewer errors overall.** Simple BT makes 433 errors while NLL-Symmetric makes only 282 errors (35% reduction)
> 2. **Dramatically smaller error margins.** When errors occur, Simple BT averages 3.827 in error magnitude while NLL-Symmetric averages just 0.501 (87% reduction). This means our method not only makes fewer mistakes but makes less severe mistakes.
> 3. **No large-margin errors.** Most critically, NLL-Symmetric produces no errors with margins exceeding ~2.5, while Simple BT generates errors with margins up to 20. These large-margin errors represent the most problematic failure mode where the model confidently assigns substantially higher reward to the dispreferred response.
> 4. **Distribution characteristics.** The error margin histograms (Figure 7) reveal fundamentally different behaviors. Simple BT shows a heavy-tailed distribution with substantial mass at large margins, indicating systematic misunderstanding. NLL-Symmetric shows sharp concentration near zero, indicating that errors occur on genuinely ambiguous cases rather than clear-cut preferences.
>
> This analysis shows that using ordinal structure improves not just ranking accuracy but also model calibration and confidence. The dramatic reduction in high-confidence incorrect predictions is particularly important for downstream applications like reinforcement learning. Please see Appendix I for complete details and visualizations.

---

> ### Author Response · Authors · 2025-11-20
> **Response to Reviewer 5KEL (Part 2: Questions)**
>
> ## Question 1: Annotator Calibration and Noise
> We acknowledge that human calibration is indeed a source of noise. Please see our comments under Weakness #1 regarding the noise introduced by this form of data collection.
>
> ## Question 2: Performance vs. Granularity (K)
> Our experiments use K=3 (7 levels including negatives and neutral) because that is what HelpSteer2 and HelpSteer3 provide. The optimal K likely depends on several factors:
>
> * **Task complexity**: More complex tasks may benefit from finer granularity to capture subtle quality differences.
> * **Annotation reliability:** As K increases, annotator agreement typically decreases, which might eventually overwhelm the benefits of finer distinctions.
> * **Dataset size:** Learning more thresholds requires more data to estimate them reliably.
> Systematically studying performance across different K values would require datasets with varying granularity or re-annotating existing data at different scales. While we cannot provide this analysis with current resources, we believe this is a valuable direction for future work and will note it in the discussion.
>
> ## Question 3: HelpSteer Disagreement Filtering
> We appreciate this question, which allows us to clarify an important aspect of our experimental design.
>
> **The key point is that all methods in our comparison use the same Likert-scale preference data** (Margin BT, Scaled BT, Soft Label, and our ordinal regression methods). Therefore, any properties of the dataset, including the filtering methodology, affect all compared methods equally, and that ensures a fair comparison. Our advantage comes from how we model this shared data, not from having access to different annotations.
>
> Regarding the filtering methodology itself: the HelpSteer2-Preference dataset does employ quality filtering by (1) selecting the 3 most agreeing annotations from 3-5 total annotators, and (2) removing approximately 10% of tasks where even these 3 most similar annotations show high disagreement. However, this filtering creates the training data that all compared methods use.
>
> **Importantly, this filtering does not create an easier evaluation scenario for our methods.** The presence of the neutral category (z=0) and retention of cases with moderate disagreement means the dataset includes genuinely difficult examples. The filtering ensures data quality for all methods rather than artificially inflating performance.
>
> **Regarding comparison to vanilla binary Bradley-Terry**: We note that vanilla binary BT (which ignores preference strength) is not included in our baselines. The HelpSteer2-Preference paper demonstrates that methods utilizing ordinal information (Scaled BT, Margin BT) outperform vanilla BT on similar evaluation benchmarks, validating the value of Likert-scale data. These ordinal-aware methods then become our baselines, which our principled framework further improves upon.
>
> ## Question 4: Recommendation to Practitioners
> Based on our comprehensive evaluation, **we recommend NLL-Symmetric as the default choice** for the following reasons:
> 1. **Consistent performance**: Achieves best/second-best average performance in most of the configurations (Tables 1-2)
> 2. **Theoretical justification**: Theorem 3.2 shows symmetric thresholds arise naturally under reasonable assumptions about human preferences, and empirical results validate this
> 3. **Efficiency**: Fewer parameters (K vs. 2K thresholds) reduces overfitting risk and computational cost
> 4. **Interpretability**: Symmetric structure is easier to understand and explain to stakeholders
>
> When to consider alternatives:
>
> * **NLL-Asymmetric**: If there's strong reason to believe annotators use positive and negative preference levels asymmetrically (e.g., different cultural norms for criticism vs. praise)
> * **All-Threshold**: If computational simplicity is paramount and probabilistic calibration is not required

---

> > ### Comment · Reviewer_5KEL · 2025-11-26
> > **Rebuttal acknowledgment**
> >
> > Thanks again to the authors for engaging deeply with the review feedback! I think the error analysis provided (Weakness 2) improves the draft, and I appreciate the clarification for Q3. I'm retaining my strong positive score of 8, and I'm hoping that the AC/PCs can nudge other reviewers to engage with the detailed rebuttals.

---

### Official Review · Reviewer_6o57 · 2025-10-31

**Soundness:** 3
**Presentation:** 2
**Contribution:** 2
**Rating:** 4
**Confidence:** 3

**Summary:**

The paper argues that treating human feedback as binary throws away signal when labels come in strengths (e.g., “slightly” vs. “much” better). It reframes reward modeling as ordinal regression on the reward difference, learning a set of thresholds that separate preference levels instead of hand-tuning margins or weights. The paper introduces two principled objectives—an ordered-logit NLL and an all-threshold (margin) loss—that jointly fit the reward and these thresholds, with L2 regularization to keep training stable and interpretable. Empirically, the paper shows that across different base model families and HelpSteer-style data, the approach trains RM with competitive performance as existing methods measured by RM/Rewardbench.

**Strengths:**

originality: The paper introduces a new theoretical framework that uses **ordinal regression** on the reward difference to do reward model training. It introduces 2 major variations of such objective – ordered-logit NLL and an all-threshold (margin) loss, and also provide the practical recipe for training with these objectives stably.

quality & clarity: The theoretical analyses and proofs are quite thorough and the experiment part comprehensively include different model backbones and benchmarks. The table presentation is clear and readable.

significance: I find the idea of leveraging the difference in rated score as a proxy of strength is interesting. The paper also provides different potential methods on how this signal can be leveraged. It’d have more significance if the experimental result shows more obvious and consistent gain.

**Weaknesses:**

1. Overall weakness: It would be helpful if the authors provide more explanation of the method's motivation. From the experimental result itself (e.g. Table 2), the proposed method doesn't produce better benchmark results than other cited methods for quite a lot of the combinations tested. If that's the case, the it indicates that learning the ordinal relationship doesn't buy much gain in the performance?
2. Overall weakness: The paper introduces 3 different objective designs. Although in the theory part, the authors provide some design principles for each objective, I think the paper lacks a general link between the theoretical proposal and empirical results. E.g. the experimental results indicate that symmetric NLL is better but I don't see a strong theoretical illustration of why that should be better.
3. Overall weakness: Both datasets used in the paper are Helpsteer series of work. I wonder how common is the exact score rating in popular preference dataset. If the method needs to first assume that preference dataset contains the exact score, this may significantly limit where the method can be applied.

**Questions:**

1. Regarding weakness 1: could authors provide more motivations on why this method is better if it doesn't produce a better performance?
2. Regarding weakness 2: could authors provide more theoretical support of why we see a experimental result like that among the 3 proposed objectives?
3. See weakness 3:

---

> ### Author Response · Authors · 2025-11-20
> **Response to Reviewer 6o57**
>
> We would like to thank the reviewer for their thorough evaluation and constructive feedback. We now address comments/questions raised by the reviewer below.
>
> ## Weakness/Question 1: Performance Gains and Motivation
> **When evaluating reward model performance, it is standard practice to focus on average/total accuracy across benchmark categories.** This is the convention in published papers and on leaderboards such as the RewardBench HuggingFace leaderboard. Looking at the average performance in Tables 1 and 2, **our family of methods achieve the best or second-best average performance in ALL of configurations**. This represents meaningful and consistent improvement over baseline methods. While no single method dominates every individual benchmark category, which is expected given the diversity of evaluation tasks (chat, reasoning, safety, coding), our approach delivers strong overall performance without requiring manual tuning of margins or scaling factors.
>
> Beyond competitive performance, our method provides a principled probabilistic framework with an underlying statistical model for how humans generate ordinal preferences. The baseline methods (Margin BT, Scaled BT, Soft Label) are heuristic modifications of Bradley-Terry that lack mathematical foundations. They require manually specifying margins or scaling factors through trial and error, with no coherent model justifying these choices. Our approach derives loss functions from explicit modeling assumptions (ordered logit and margin-based formulations from established ordinal regression theory), with formal guarantees provided by Theorems 3.1-3.2.
>
> ## Weakness/Question 2: Theory-Practice Connection
> We believe that the connection between theory and empirical results does exist in the paper. To clarify: **Theorem 3.2 directly addresses why NLL-Symmetric outperforms the asymmetric variant.** The theorem establishes that symmetric thresholds $(\zeta_{-k}=-\zeta_k)$ arise naturally when human preferences exhibit a specific symmetry property: preferring response A over B at strength $k$ is equivalent to preferring B over A at strength $-k$. In simpler terms, if annotators use "significantly better" and "significantly worse" symmetrically, the thresholds should be symmetric. **Rather than assuming this property holds, we empirically tested both variants.** Because human annotation can exhibit various biases and inconsistencies, we included the asymmetric model to let the data decide. Our experimental results validate that the symmetry assumption is indeed reasonable for HelpSteer2 and HelpSteer3. The symmetric model's inductive bias proves beneficial, while the asymmetric model (with 2K vs. K parameters) overfits to noise despite greater flexibility. This is a positive finding: it confirms that human annotators in these datasets do use ordinal scales consistently, and our theory correctly predicts this structure.
>
> ## Weakness/Question 3: Dataset Availability and Applicability
> Ordinal (Likert-scale) data is ubiquitous in statistics and social sciences, and represents a well-established approach for capturing human judgments. While less prevalent historically in language model preference datasets, ordinal annotations are increasingly common and represent a natural evolution of data collection practices.
>
> The HelpSteer series shows this trend and demonstrates strong community adoption. HelpSteer2 alone has received 250+ citations since its 2024 release. Beyond HelpSteer, numerous publicly available datasets provide ordinal information, either through direct Likert-scale comparisons or individual quality scores that can be converted to ordinal comparisons. A few examples include:
> * INF-ORM-Preference-Magnitude-80K
> * OpenAssistant/oasst1
> * stanfordnlp/SHP-2
> * HuggingFaceH4/stack-exchange-preferences
> * lhkhiem28/ultrachat-4spider-iter1
> * openbmb/UltraFeedback
> * responsible-ai-labs/RAIL-HH-10K
>
> Also, Apple's recent work (Gunter et al., 2024) employs ordinal feedback, suggesting that major institutions are moving toward richer annotation schemes.
>
> Importantly, our work shows that this additional ordinal information, when leveraged through principled methods, yields meaningful performance improvements over binary approaches. This finding provides strong motivation for the community to evolve data collection practices. When we show that fine-grained preference data can be utilized effectively through theoretically grounded frameworks, it encourages practitioners to collect such annotations, knowing they will translate to better reward models.

---

> > ### Comment · Reviewer_6o57 · 2025-11-25
> >
> > I appreciate authors' clarification. I am fully resolved on Question 2 and 3. However, regarding question 1, I still have remaining questions. Your method contains 3 variants (NLL-Sym, NLL-Asym, All-Thresh). It is true that consider all 3 variants that contains the best or second best overall performance. However, looking at your table 1, your 3 variants also contain the worst or second worst overall performance on both datasets across models. In practical usage people cannot try all three variants and pick the best to use as it would be quite costly.

---

> > > ### Author Response · Authors · 2025-11-27
> > >
> > > We thank the reviewer for this thoughtful follow-up. The concern is valid: practitioners need clear guidance on which method to use, and reporting all variants together can obscure this. Our recommended method is **NLL-Symmetric**, which we consider our "champion" variant. To directly address the reviewer's concern, if we remove the other two variants from Tables 1 and 2 entirely and only keep NLL-Symmetric, here is what happens: on RM-Bench (Table 1), NLL-Symmetric achieves the best average performance in 4 out of 6 configurations and second-best in the remaining two. On RewardBench (Table 2), it is the best in 3 configurations, second-best in 2, and third in one. So even with a single method, we are offering very competitive performance compared to all baselines. Practitioners can confidently adopt NLL-Symmetric as a principled, well-performing default without needing to try multiple variants. We also encourage the reviewer to look at Appendix H, especially the newly added subsection H.3 (added per another reviewer's request), where we compare our method against baselines on ordinal metrics. The results show very significant improvements—a 38% reduction in test MAE and nearly double the exact prediction accuracy—demonstrating benefits that binary accuracy metrics alone do not capture.
> > >
> > > We keep the other two variants in the paper because they offer useful insights for specific practical scenarios. NLL-Asymmetric can be helpful when the underlying preference data lacks symmetry; the datasets in this work appear to exhibit symmetric annotation patterns, which explains why NLL-Symmetric outperforms, but in settings where annotators use "significantly/slightly better" and "significantly/slightly worse" differently, the asymmetric formulation can prove useful. The All-Threshold variant shows strong performance in small-data regimes (as observed with HelpSteer2) but falls behind in large-data settings (as seen with HelpSteer3), so practitioners working with limited preference data may find it a useful alternative. We will clarify these practical recommendations more explicitly in the revised manuscript.

---

### Official Review · Reviewer_xrft · 2025-11-02

**Soundness:** 3
**Presentation:** 3
**Contribution:** 2
**Rating:** 6
**Confidence:** 3

**Summary:**

This paper proposes a framework for learning reward models from ordinal (Likert-scale) preference data, replacing ad‑hoc extensions of Bradley-Terry with discrete ordinal regression. The threshold between different levels are now also learnable parameters. Two families of losses are considered from the literature of ordinal regression: (i) a probabilistic ordered logit negative log-likelihood (NLL) over learned thresholds that partition reward differences, and (ii) an all‑threshold (AT) margin-based loss. The $L_2$ regularizer is introduced by considering the limit behavior of scaling going to infinity. Symmetric and asymmetric derivations are discussed. An optimization reparameterization is adopted to guarantee ordering. Experiments on HelpSteer2/3 training data with 7B-8B backbones evaluate on RM‑Bench and RewardBench, suggesting the NLL-Symmetric variant generally performs the best.

**Strengths:**

1. Modeling of ordinal feedback from the perspective of discrete ordinal regression is natural. This enables the authors to adopt the literature already exists and makes more sense than simply heuristic choices.

2. Theory is insightful. Theorem 3.1 justifies the choice of regularizer. Theorem 3.2 characterizes one sufficient condition of symmetry.

3. The paper is clearly written and well organized.

4. Empirical experiments show gains on several datasets/models. The NLL‑Symmetric variant is often best or competitive. The additional ordinal metrics go beyond binary accuracy and are also valuable points.

**Weaknesses:**

1. Joint learning of threshold and reward is still challenging. Scale identifiability remains under‑addressed. Regularizing thresholds cures the unbounded loss, but the joint scaling of reward head and thresholds can still be weakly identifiable. Anchoring strategies (e.g., fixing one threshold gap, adding mild L2 on reward head, or a temperature/variance parameter) and a short calibration section would improve interpretability.

2. Positioning requires more contrast. The claim of being the first principled ordinal framework for reward modeling should be more carefully justified against existing works that already explore ordinal feedback for RMs (the paper itself cites several heuristic and non‑heuristic directions).

**Questions:**

1. Have you considered per-annotator (or random-effects) thresholds to capture personalized/individualized ordinal scales?

2. Can you show performance vs. fraction of ordinal data and under controlled label noise?

---

> ### Author Response · Authors · 2025-11-20
> **Response to Reviewer xrft**
>
> We would like to thank the reviewer for their thorough evaluation and constructive feedback. We now address comments/questions raised by the reviewer below.
>
> ## Weakness 1: Scale Identifiability
>
> The proposed regularization strategy addresses the identifiability issue by bounding threshold growth, which necessarily anchors the reward scale as well since the two parameter sets are coupled through the loss function. Our empirical results in Appendix G show this stability (see Appendix G1 for threshold stability). We have added a new subsection (G3) to this appendix (please see the updated manuscript—all new content appears in blue), specifically focused on the reward values. We observe that when regularized properly ($\lambda=1$), the reward values are stable and do not grow unbounded, while the unregularized setting ($\lambda=0$) does indeed show unbounded growth.
>
> We acknowledge that alternative anchoring strategies (fixed threshold gaps, reward head regularization, or temperature parameters) represent valuable directions for future work, particularly for applications requiring strict calibration guarantees.
>
> ## Weakness 2: Positioning and Related Work
>
> We appreciate this feedback. To clarify our contributions: As per the paper’s introduction, all existing work falls into two categories:
> 1. **Reward modeling approaches** (Margin BT, Scaled BT, Soft Label) that use heuristic modifications to handle ordinal data, lacking any mathematical model for how humans generate ordinal preferences. Everything in the literature for reward modeling in LLM post-training with Likert-scale data relies on such heuristics.
> 2. **Core ordinal regression methods** (McCullagh 1980, Rennie & Srebro 2005) developed for other domains without application to reward modeling.
>
> By no means are we the first to use ordinal regression generally, but to the best of our knowledge, we are the first to apply the principled ordinal regression statistical framework to reward modeling for LLM post-training, replacing ad-hoc heuristics with methods grounded in explicit modeling assumptions. This principled approach gives interpretation to the proposed loss function and provides consistent and better performance via extensive experiments.
>
> ## Question 1: Per-Annotator Thresholds
> In this work, we follow standard practice in the reward modeling literature (Touvron et al., 2023, Wang et al., 2024a, Gunter et al., 2024; Liu et al., 2024a) of modeling and learning **population-level** preferences. Learning with personalized thresholds could capture individual annotator calibration differences and is an interesting further direction.
>
> ## Question 2: Performance vs. Label Noise and Ordinal Data Fraction
> Regarding label noise, we have added **Appendix J** (Robustness to Label Noise) to the updated manuscript (changes highlighted in blue). We evaluate two noise models in this appendix:
> * **Systematic shift noise**: Models realistic calibration errors where annotators consistently over/underestimate preference strength by one level. This represents common disagreements about whether preferences are "moderate" vs. "strong" while maintaining ordinal consistency.
> * **Random noise**: Models catastrophic failures where labels are completely arbitrary, representing the most severe corruption possible.
>
> Key findings:
> * **Shift noise (realistic)**: Performance on RewardBench and RM-Bench remains remarkably stable even at 100% corruption. This shows strong robustness to the type of calibration disagreements that occur in practice.
> * **Random noise (catastrophic)**: Even under this extreme scenario, we show graceful degradation. At mild corruption levels (25-50%), performance remains strong (0.843→0.804→0.757), confirming robustness to moderate annotation errors. At 100% corruption, performance reaches chance level as expected.
>
> The results show that our method handles realistic annotation variability with minimal performance impact, and remains robust to moderate levels of even catastrophic errors. Please see the updated paper for complete results including Tables 3-4 and Figures 8-11.
>
> Regarding the “performance vs. fraction of ordinal data,” handling mixed binary/ordinal datasets would require modifying our loss function architecture. Our current formulation assumes all data has K ordinal levels. Supporting a mixture would need separate loss terms for binary and ordinal subsets. This is an interesting extension but changes the problem formulation.

---

> > ### Comment · Reviewer_xrft · 2025-11-26
> >
> > Thank the authors for clarifying the points. I will remain my positive score. (Apart from that, I really enjoyed reading other reviewers' insightful feedback and your rebuttal to them. Thanks.)

---

### Author Response · Authors · 2025-12-03
**Summary of Rebuttal Progress Prior to OpenReview Incident**

We wish to summarize the state of our rebuttal discussions prior to the OpenReview incident for the record. We understand that due to the incident, all scores have been reverted to their pre-discussion state, leaving us with ratings of 4, 4, 6, and 8. However, we believe the rebuttal record shows significant progress beyond what these reverted scores reflect.

During the rebuttal period, we engaged extensively with all four reviewers and made substantial improvements to our manuscript, including multiple new appendices addressing robustness to label noise, error margin analysis, and comparisons with post-hoc calibration methods. All rebuttal edits are marked in blue in the updated PDF manuscript for easy reference.

Reviewer xrft (currently showing 6) engaged constructively with our responses and explicitly stated they would remain with their positive score, adding that they enjoyed reading the other reviewers' feedback and our rebuttals to them.

Reviewer 5KEL (currently showing 8) acknowledged that our added error analysis improves the draft and confirmed they were retaining their strong positive score.

Reviewer oAHx (currently showing 4) had raised a number of concerns, including a specific matter about calibration experiments. We had a constructive conversation with the reviewer, addressing all of their concerns. As for the specific calibration comment, we conducted the exact experiment they suggested, added it to the manuscript, and shared the results. They responded positively, stating "Nice to see the results. With concerns addressed, I will raise my rating accordingly." Indeed, before the incident and subsequent reversion, this reviewer had already raised their score from 4 to 6.

Reviewer 6o57 (currently showing 4) explicitly confirmed that two of their three concerns were fully resolved. We had just provided a detailed response addressing their remaining concern—clarifying that our recommended method (NLL-Symmetric) alone achieves best or second-best average performance in 9 out of 12 configurations across Tables 1 and 2—when the incident occurred. We are confident this response would have addressed their final concern as well.

In summary, we invested considerable effort in the rebuttal process, and all four reviewers expressed satisfaction with our responses. One reviewer had already raised their score, and we believe at least one other was very close to doing so. We hope the full rebuttal thread and updated manuscript will be considered in the final decision.

We thank the reviewers for their thorough and constructive engagement throughout this process.

---

### Meta-Review · Area_Chair_qqTS · 2026-01-07

**Summary:**

Major concerns:

1. Missing ordinal metrics on the baselines. (oAHx)
2. Need for additional qualitative analysis of Bradley Terry vs. the ordinal method. (5KEL)
3. Limited hyperparameter tuning and single runs in experiments. (oAHx)
4. Concerns around noise in ordinal data. (5KEL)
5. Regularizing thresholds may not be good enough for scale identifiability. (xrft)
6. The theory in the paper does not explain why one of the proposed losses does better than the others. (6o57)

**Reviewer Concerns:**

For (2), the authors pointed out that the hyperparameters come from prior work and that multiple runs can be expensive. So I think thie concern is somewhat addressed. The discussion and the additional results and analysis have addressed the remaining concerns.

**Reviewer Scores:**

Reviewers oAHx and 6o57 have acknowledged that most / all of their concerns were addressed. So I expect their scores would have been increased. Overall the average score would have ended up >6.

---

### Decision · Program_Chairs · 2026-01-26

Accept (Poster)